# Moisture availability and groundwater recharge paced by orbital forcing over the past 750,000 years in the southwestern USA
Simon D. Steidle[1] ✉, Kathleen A. Wendt[2], Yuri Dublyansky [1], R. Lawrence Edwards[3], Xianglei Li [3,4], Gracelyn McClure[3], Gina E. Moseley [1] & Christoph Spötl [1]

Quaternary climate changes are driven in part by variations in the distribution and strength of insolation due to orbital parameters. Continental climate variability is well documented for the most recent glacial-interglacial cycles, yet few records extend further back in time. Such records are critically needed to comprehensively assess the entire spectrum of natural climate variability against the backdrop of anthropogenic warming. Here, we apply uranium isotope geochronology to calcite deposits to date groundwater-table changes in Devils Hole cave, Nevada. The deposits record multi-meter groundwater-table fluctuations over the last 750,000 years, reflecting the long-term evolution of hydroclimate in this presently arid region. During periods between glacial or interglacial extremes, the water table responded sensitively to variations in 65°N summer insolation, likely caused by the increasing extent of North American ice sheets during cold period, which steered moisture-laden trajectories towards the southwestern USA. These orbitally-driven hydroclimatic changes are superimposed on a tectonically-driven long-term decline in the regional groundwater table observed prior to 438,000 ± 14,000 years ago.

Anthropogenically forced climate change poses a significant threat to groundwater resources and exacerbates socio-environmental conflicts in the southwestern USA[1–5]. Radiometrically dated hydroclimate archives that extend over multiple glacial cycles provide important quantitative records of water availability under different climate regimes, thus enabling a deeper understanding of the driving mechanisms and tele-connections. Records of Antarctic ice cores (e.g., ref. 6) and ocean sediments (e.g., ref. 7) provide foundational information on past climate change during glacial-interglacial cycles. However, the oldest portion of these records are limited with respect to the accuracy of their chronologies, because orbital tuning techniques introduce circular reasoning when attempting to validate insolation as the driving mechanism of global climate. Furthermore, both ice cores and ocean sediments do not directly capture the climate dynamics on midlatitude continents. Instead, we turn to terrestrial archives, such as speleothems, that can be precisely and accurately dated using U-series techniques[8]. A common limitation of speleothem archives is their short temporal coverage. This can be

circumnavigated by splicing individual speleothem records to cover several glacial-interglacial cycles (e.g., ref. 9).

In this study, we use continuously deposited phreatic speleothems from Devils Hole, Nevada, to track groundwater table changes over eight glacial-interglacial cycles spanning the time since 750 ka (ka stands for thousand years before present and refers to a point in time), providing the longest record of moisture availability and groundwater recharge in this water-limited region of North America.

During the Quaternary, the southwestern region of North America repeatedly switched between arid, warm interglacial climates, as are characteristic of the region today, and humid, cooler glacial climates that led to the filling of extensive lakes and higher groundwater tables[10–13]. Increased moisture availability was closely tied to the expansion of the Laurentide ice sheet[14,15] due to the presence of anticyclones that aided in redirecting winter moisture from the northern Pacific further south into the Great Basin[15,16]. Under contemporary conditions, winter precipitation remains the primary source of regional moisture in the Great Basin, with the summer monsoon

[1]Institute of Geology, University of Innsbruck, Innsbruck, Austria. [2]College of Earth, Ocean, and Atmospheric Sciences, Oregon State University, Corvallis, OR, USA. [3]School of Earth and Environmental Sciences, University of Minnesota, Minneapolis, MN, USA. [4]Institute of Vertebrate Paleontology and Paleoanthropology, Chinese Academy of Science, Beijing, China. ✉e-mail: simonsteidlescience@gmail.com

contributing only a minor portion (<10%) to annual local recharge[17]. It has been shown that the summer monsoon was even further suppressed during glacial periods[18,19].

Devils Hole, situated in the Ash Meadows Groundwater Basin of southern Nevada (Fig. 1), is positioned within the hydrologically enclosed Great Basin. This subvertical extensional fracture intersects the groundwater table, enabling large-scale changes in moisture availability and groundwater chemistry which are related to atmospheric changes to be preserved in submerged calcite deposits. These deposits, precipitated from groundwater onto the cave walls, have been shown to preserve a water-table elevation history[11,20,21]. Due to its proximity to the Yucca Mountain nuclear waste repository, the paleo water-table history of this region has been a topic of study since the 1980s. A pioneering study [22] reported calcite veins, dated to 750 ± 50 ka, perched at +26 m above the present-day water table on the surface of the Devils Hole ridge. Using this and two younger samples, these authors calculated a mean rate of water-table decline of 2–3 m/100 kyr (kyr stands for thousands of years and refers to a duration) over the past 750 kyr. They also noted a more rapid decline of about 8 m/100 kyr around 700 ka, followed by a slower decline of 2 m/100 kyr over the last 510 kyr. Tectonic uplift was proposed as the primary driver for these long-term trends[22]. Subsequent high-resolution studies at Devils Hole cave revealed large ( + 10 m) paleo water-table oscillations that occurred in step with the last three glacial-interglacial cycles[11,20]. Superimposed on these glacial-interglacial changes are millennial-scale highstands that coincide with North Atlantic climate variability during the last three glacial cycles[11]. Climate-induced highstands are associated with up to +240% increase in groundwater recharge, as simulated by a regional groundwater-flow model[23]. The Devils Hole water table over the last 350 kyr show a clear link to regional hydroclimate change and suggests little to no long-term water-table decline, in contrast to Winograd and Szabo[22]. Here, we extend the Devils Hole paleo water-table record to examine the climate-induced changes while reconciling the long-term decline during the mid to late Quaternary.

The large water-table fluctuations over glacial-interglacial cycles at Devils Hole are also paralleled by distinctive geochemical characteristics of

the calcite[21,22,24,25]. Specifically, the ratio of uranium isotopes recorded in the subaqueous calcite offers a qualitative link to the aquifer's water-table changes[25] in addition to their date of formation[26]. Calcite deposits found in caves are commonly dated using the $^{230}$Th–U method, where the radioactive decay of $^{238}$U and $^{234}$U results in the ingrowth of $^{230}$Th. Through an iterative process, an initial ratio of $^{234}$U/$^{238}$U ($\delta^{234}U_{initial}$) is estimated and the age ($^{230}$Th–U age) of formation is calculated[27]. The short-lived $^{230}$Th isotope limits the applicability of this method to samples younger than about 650 ka[8]. At Devils Hole, calcite deposits extend beyond this limit (up to 820 ka), offering an exceptional opportunity to investigate a terrestrial hydroclimate record covering eight glacial-interglacial cycles. Establishing an absolute-dated chronology beyond 650 ka is possible with $^{234}$U–U dating methods, which rely on a robust understanding of past $\delta^{234}U_{initial}$ values. The Devils Hole record shows a relatively small variation in $\delta^{234}U_{initial}$ values (1650–1850‰) over the last 650 ka[26]. $\delta^{234}U_{initial}$ values at Devils Hole can be further constrained by statistically significant correlations with $\delta^{18}O$ and $\delta^{13}C$ derived from the same calcite samples[26]. As a result, $^{234}$U–U ages can be determined with relative age uncertainties of a few percent. $^{234}$U–U dating has been shown to be a viable method for determining the age of Devils Hole calcite that is older than the limit of $^{230}$Th–U dating[26].

## Results and discussion
### $^{230}$Th–U and $^{234}$U–U chronologies
For each measured sample, there are two possible approaches for calculating an age. $^{230}$Th–U ages[27] can be used for samples up to 650 ka, which is the approximate limit of $^{230}$Th–U dating (Fig. 2). For Devils Hole, the relationship between $\delta^{234}U_{initial}$ and $\delta^{13}C$ and $\delta^{18}O$[26] was used to also calculate $^{234}$U–U ages based on the decay of $^{234}$U. In total, 92 samples were processed in the range of 350–820 ka and five outliers were removed (see Supplementary Material). For 59 samples where both ages could be derived (i.e., they are within the physical limits of $^{230}$Th–U dating, defined here by a $^{230}$Th–U age uncertainty <1 million years; Fig. 2A), a bootstrap analysis (100,000 iterations) of their linear relation was performed. For a linear correlation ($^{234}$U–U age = a + b * $^{230}$Th–U age) the analysis yielded the

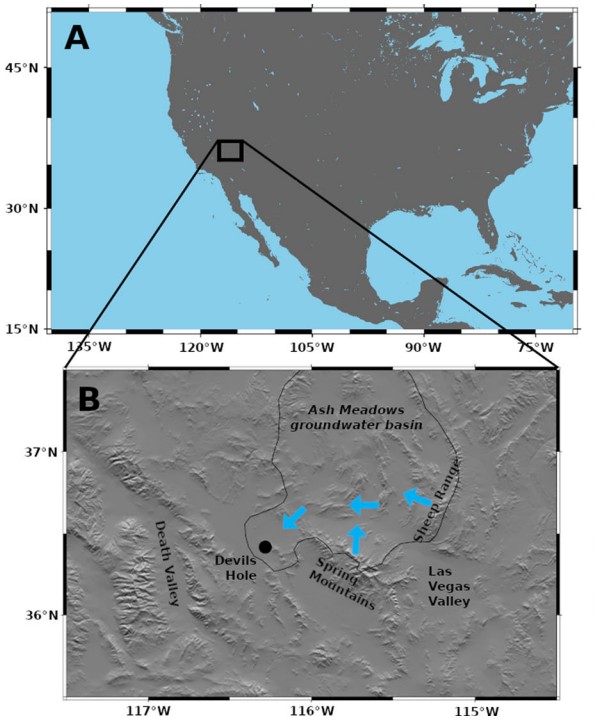

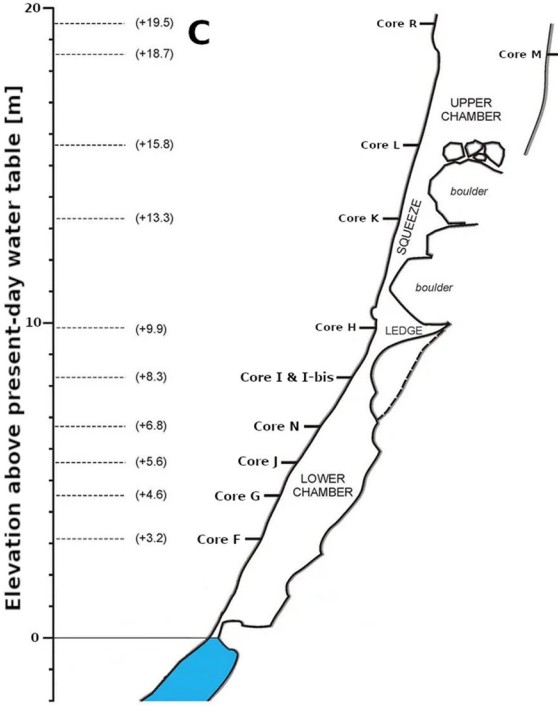

**Fig. 1 | Region and cave map. A** Overview map of North America. **B** Shaded relief map of the study area showing the Ash Meadows Groundwater Basin, Devils Hole and the two main regions of groundwater recharge (Spring Mountains and Sheep

Range). Blue arrows indicate the main direction of groundwater flow. **C** Simplified transect of Devils Hole #2 with the position of horizontally drilled cores in the hanging wall of this fracture, providing a long-term record of calcite deposition.

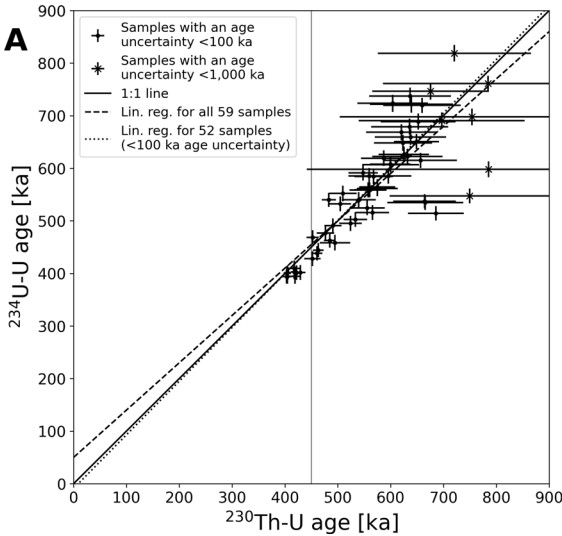

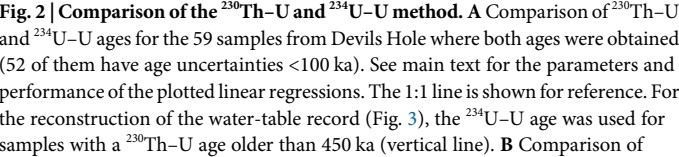

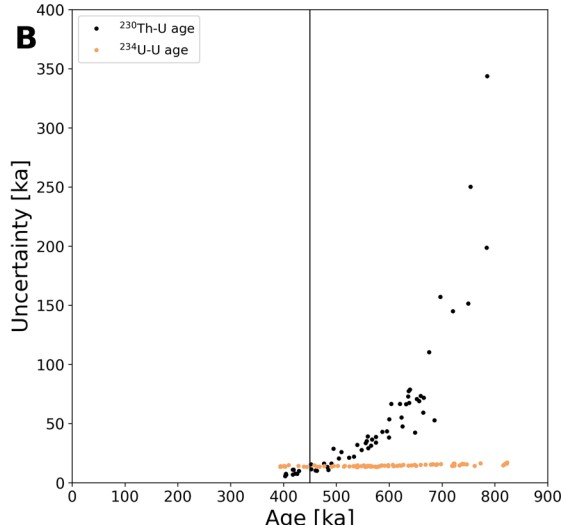

**Fig. 2 | Comparison of the $^{230}$Th–U and $^{234}$U–U method. A** Comparison of $^{230}$Th–U and $^{234}$U–U ages for the 59 samples from Devils Hole where both ages were obtained (52 of them have age uncertainties <100 ka). See main text for the parameters and performance of the plotted linear regressions. The 1:1 line is shown for reference. For the reconstruction of the water-table record (Fig. 3), the $^{234}$U–U age was used for samples with a $^{230}$Th–U age older than 450 ka (vertical line). **B** Comparison of uncertainties of 60 $^{230}$Th–U and 86 $^{234}$U–U ages (see Supplementary Material for the selection of samples). $^{230}$Th–U ages older than 450 ka (vertical line) were not used for the water-table reconstruction (Fig. 3) and were substituted by their $^{234}$U–U age. All age uncertainties are 2σ.

following parameters: $a = 50 \pm 101$ ka, $b = 0.90 \pm 0.20$, $R^2 = 0.71 \pm 0.16$ and $P$ value = 5.5E-10 ± 1.8.0E-07 (uncertainties are 2σ standard deviations). The $P$ value reflects the likeliness of the null hypothesis of no concordance. Because $^{230}$Th–U ages close to secular equilibrium scatter towards exceedingly high numbers (an effect not balanced by equal scatter at low values), we repeated the analysis for samples with an uncertainty of <100 ka for their $^{230}$Th–U age ($n = 52$). This analysis yielded the following parameters: $a = −9 \pm 99$ ka, $b = 1.02 \pm 0.20$, $R^2 = 0.76 \pm 0.17$ and $P$ value = 2.9E-10 ± 3.9E-08. Overall, the results of $^{230}$Th–U and $^{234}$U–U dating show good agreement (Fig. 2).

The quoted age uncertainties incorporate the statistical uncertainty in the mass spectrometric measurement and a systematic uncertainty related to the estimate of the initial $^{230}$Th. For the calculation of $^{230}$Th–U ages, the statistical uncertainty is apparent in the uncertainty of the "uncorrected Th-age" and a systematic uncertainty contribution from detrital $^{230}$Th is mostly negligible, owing to the age of the samples and the low concentrations of $^{232}$Th, which serves as a proxy for detrital $^{230}$Th (measured atomic ratio $^{230}$Th/$^{232}$Th > $9.91 \times 10^{−4}$). For the calculation of $^{234}$U–U ages, the statistical uncertainty is sourced in the measured $\delta^{234}$U, which is small compared to the systematic uncertainty in estimating $\delta^{234}$U$_{initial}$. The 2σ uncertainty of $\delta^{234}$U$_{initial}$ in Devils Hole calcite was calculated[26] to be 60.5‰ for all samples. For 78 samples (85%) $^{230}$Th–U and $^{234}$U–U ages agree within their 2σ uncertainties and only four samples (4%) have a difference of more than 1.5× their combined uncertainty.

Samples deposited around 450 ka display a similar uncertainty for both age calculation methods (Fig. 2B). For the purpose of reconstructing the water-table record, all samples with a $^{230}$Th–U age >450 ka are plotted using their $^{234}$U–U age.

**Construction of water-table changes**
The oldest mammillary calcite deposition is recorded in four cores between +4.6 m and +8.3 m elevation and dates to 820 ka. This value is the mean of samples Ibis825, N771, N775, J992, and G867. Its uncertainty is limited by the $^{234}$U–U age model and is about ±15 ka. Apart from this 820 ka layer, the next youngest mammillary calcite formed around 750 ka and is present at all sampled elevations above 4 m. The water-table decline after the brief rise at 746 ± 16 ka (+19.5 m) marks the start of the continuous record of the water-table changes (Fig. 3).

Water-table oscillations prior to 350 ka reached up to 15 m in amplitude, and their rate was comparable to the rate of water-table drops during Terminations in the last 350 kyr[11,20]. Furthermore, our record reveals smaller amplitude oscillations in between which show periods of roughly 20 kyr. Mammillary calcite deposition at the highest elevations (>15 m) only occurred prior to 600 ka, while mammillary calcite layers in cores at 8–10 m elevation become thinner in more recent glacial periods (cf. longer and more frequent periods of submergence at these elevations in the older part as shown in Fig. 3).

**Long-term water-table decline component**
The youngest ages of mammillary calcite obtained from each of the five highest cores become older as the elevation of the core increases: at +19.5 m the last deposition dates to 719 ± 14 ka, at +18.7 m to 660 ± 15 ka, at +15.8 m to 643 ± 15 ka, at +13.3 m to 567 ± 14 ka. The core at +9.9 m contains a last thick (>1 cm) mammillary calcite layer that formed at 438 ± 14 ka. Four out of these five mammillary calcite layers (all but the one at +15.8 m) are overgrown by folia or proto-folia indicating a water-table decline immediately after their deposition. Considering an age of 719 ka at +19.5 m and 438 ka at +9.9 m suggests a mean rate of water-table decline of 3.4 m per 100 kyr on which shorter (<100 kyr) water-table variations are superimposed. The end of calcite deposition at elevations of +18.7 m, +15.8 m and +13.3 m is roughly consistent with a long-term linear component of a declining water table over this time period (Fig. 3). Such a long-term trend is not apparent after 438 ± 14 ka[11].

**Lower limit of water-table changes**
The identification of long (>100 kyr) periods of continuous growth of mammillary calcite provides a lower limit of water-table lowstands. Although the absence of folia may be considered an indicator of continuous submergence, folia may not necessarily form when the water-table drops or rises quickly. Another indication of continuous growth (i.e., the absence of significant hiatuses) is a constant mean growth rate of 0.9 ± 0.3 mm/kyr found in a previous study of mammillary calcite from an elevation of +1.8 m[26].

The core at +5.6 m is interpreted to have been continuously submerged from 682 ± 16 ka where we have a marker for a rising water table in the core above until 555 ± 13 ka (see Supplementary Material

**Fig. 3 | Devils Hole water-table record.** Elevation of the paleo water table in Devils Hole relative to today's water-table position (0 m), based on this study, refs. 11, and [20]. The black dashed line is the long-term component of decline discussed in the subsection "Long-term water-table decline component". Our interpretation of water-table changes across the last 750 kyr (based on refs. 20 and [11]) and this study is shown by the blue shading and gray line. Dense blue color means "submerged with high certainty", white means "not submerged with high certainty" and intermediate colors are a qualitative measure of the likeliness of submergence. They gray line is a spline function showing a likely history of the water table. Intermediate blue shading reflects lack of data, dating uncertainty and the fact that due to the slow growth of mammillary calcite, changes on timescales <10 kyr cannot be resolved. No interpretation of the water table is provided for the oldest part of the record prior to 750 ka due to the scarcity of the data. All age uncertainties are 2σ.

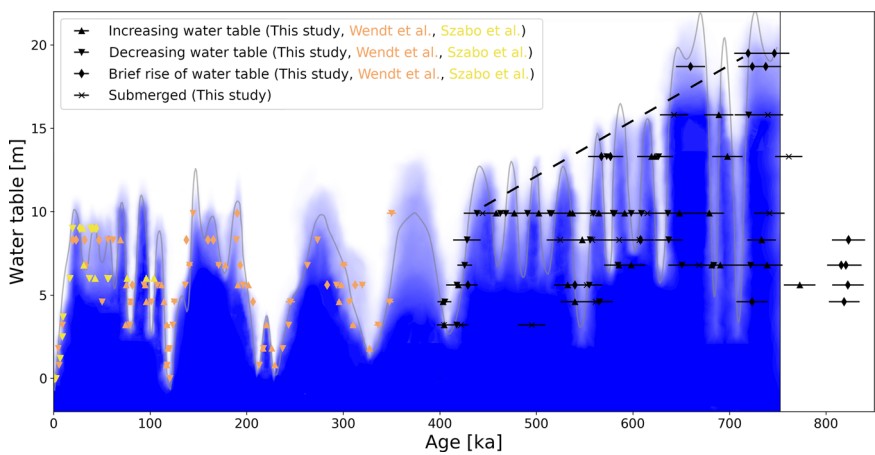

Figs. S2 and S4). This proto-folia layer was dated in the core above ( + 6.8 m) and demonstrates submergence after 682 ± 16 ka. The average growth rate of 0.8 mm/kyr as calculated agrees with the previously published rate of 0.9 ± 0.3 mm/kyr[26]. Another period of continuous growth at +5.6 m elevation lasted from 532 ± 14 ka (J784) to 418 ± 11 ka (J664). Its average growth rate of 1.1 mm/kyr also agrees with the published growth rate[26]. Both growth layers are, within their uncertainties, consistent with sampled deposits below this elevation (i.e., there was no folia deposition below +5.6 m during these two intervals).

### Limitations

Since the boundary between folia and mammillary calcite cannot be directly dated, ages for declining or increasing water table, as derived from mammillary calcite a few mm distant from the respective boundaries, are systematically too old or too young by up to 5 kyr, respectively. In this study, no attempt was made to extrapolate the age to the boundary between mammillary calcite and folia by measuring multiple ages (cf. ref. 11) because the uncertainties of these older parts of the deposit precluded such an approach. Sampling-derived offsets, however, are negligible compared to other uncertainties and do not affect the interpretation.

Given the dating uncertainty it is not possible to resolve changes that occur on timescales smaller than 10 kyr. The blue shading and gray line in Fig. 3 is our interpretation of changes on timescales >10 kyr. Some of the cores show many oscillations between thin layers of mammillary calcite and folia, suggesting higher-frequency water-table fluctuations than shown in Fig. 3. Such high-frequency fluctuations are especially prevalent during the 30–40 kyr long highstands ( + 18.7 m and +19.5 m) at 660 ka and 740 ka.

The assumption that the driving factor of $\delta^{234}U_{initial}$ and its correlation with $\delta^{18}O$ and $\delta^{13}C$ (i.e., the findings of ref. 26 and ref. 25) were constant since 750 ka is a critical aspect of the age model. Li et al.[26] provided the $\delta^{234}U_{initial}$ regression for the $^{234}U$–U age model up to 590 ka. Furthermore, Li et al.[26] considered three independent groupings of data for different time intervals. There was no significant difference found between these time intervals, providing evidence that there was no gradual change over the last 590 kyr and that the mechanism described in ref. 25 therefore was of similar significance before 450 ka. We consider it likely that there was also no gradual change for the 160 kry before that (i.e., between 750 and 590 ka), as the most likely cause for a significant change in $\delta^{234}U_{initial}$ would be an unusually strong tectonic event that caused a massive perturbation of the aquifer (i.e., a major change of the flow path(s) or the recharge area). The data provided by Perouse and Wernicke[28] do not provide evidence of an outstanding tectonic event in this time range, but rather suggest many small events which were not big enough to significantly alter the uranium

concentration integrated over the 80 km long flow path from the source region to Devils Hole. Considering the size of the aquifer and the linkage of $\delta^{234}U_{initial}$ to water-table elevation it seems unlikely that a shift in $\delta^{234}U_{initial}$ without a change in water-table height occurred. The presented data show no apparent major sudden change in this time range (750–590 ka).

The data presented in this study include $^{230}Th$–U ages starting at 784 ± 199 ka (the oldest age with an uncertainty <100 ka is 686 ± 53 ka), which is close to the start of the water-table record (at 750 ka). As presented in "Results" (and Fig. 2), these data support the legitimacy of the methods used to construct $^{234}U$–U ages over the last 750 kyr.

### Water-table elevation changes in the Ash Meadows aquifer upstream of Devils Hole

Wendt et al.[25] found that $\delta^{234}U_{initial}$ variations captured by the calcite reflect water-table changes in the aquifer upstream of Devils Hole. $\delta^{234}U_{initial}$ is expected to increase simultaneously with the rise in water table following long and deep lowstands as a result of the submergence of rock (and associated water-rock interactions) previously located in the unsaturated zone[25]. Wendt et al.[25] discussed different sources of $\delta^{234}U_{initial}$ variations in Devils Hole and found processes other than the discussed effect of water-table elevation (such as changes in the groundwater source, flow rate, or flow path) to be negligible over the past 350 kyr. It is assumed here that this conclusion can be extended back to 750 ka. This assumption is directly related to the discussion of extending the $^{234}U$–U age model back in time with the same supporting arguments and limitations (see paragraph above). Under this assumption, a water-table rise after a major lowstand in Devils Hole that was not accompanied by a rise in $\delta^{234}U_{initial}$ indicates a local origin of the water-table change that did not affect the whole aquifer and was thus not linked to climate change.

In the time period between 820 and 350 ka, only one significant (i.e., uncertainty ranges of data points in the minimum do not overlap with the uncertainty range in the following maximum) interval of low $\delta^{234}U_{initial}$ dated to around 410 ka was followed by an increase in $\delta^{234}U_{initial}$ (Fig. 4). Similar rises in $\delta^{234}U_{initial}$ but smaller in amplitude than twice the uncertainty of individual data points occurred around 700 ka and 630 ka. In all three cases, they followed a pronounced lowstand of the water table at Devils Hole and it is concluded that the whole aquifer had a significantly lowered water table.

There is no significant shift in $\delta^{234}U_{initial}$ from calcite deposition at 820 ka to the start of the water-table record at 750 ka. This could indicate that there was no long-lasting lowstand during this period, but given a data gap of about 70 kyr, major changes of the water-table elevation cannot be excluded.

**Fig. 4 | Record of $\delta^{234}U_{initial}$.** $\delta^{234}U_{initial}$ values used for the $^{234}$U–U ages in this study and $\delta^{234}U_{initial}$ values from ref. 26; partly based on direct Th–U measurements) against time. Blue shading and gray line in the background is the interpreted qualitative probability of changes of the water-table height since 750 ka on timescales >10 kyr (see Fig. 3).

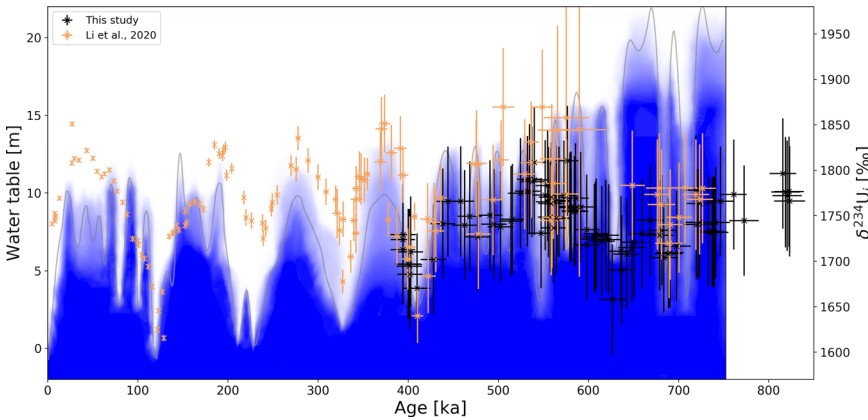

The water-table changes of lower amplitude and shorter duration between 820 ka and 350 ka are not expected to have had an influence on $\delta^{234}U_{initial}$ as such water-table changes are also not reflected in $\delta^{234}U_{initial}$ in the last 350 kyr[25].

**Water-table lowstand at 550 ka**

At 550 ± 15 ka Devils Hole experienced a major lowstand that was not accompanied by a corresponding decline in $\delta^{234}U_{initial}$. One possibility is that although being low, the lowstand was short. The markers for water-table increase and decrease indicate a duration of this lowstand of about 20 kyr but have overlapping uncertainties. It cannot be excluded that a comparably short duration was responsible for the lack of a $\delta^{234}U_{initial}$ minimum.

A second line of evidence for the regional climate around this time can be provided by stable oxygen and carbon isotopes. Glacial (MIS [=Marine Isotope Stage] 2 and MIS 6) values of about −16.7‰ in $\delta^{18}O$ and −1.7‰ in $\delta^{13}C$ and interglacial (MIS 5e) values of about −14.7‰ in $\delta^{18}O$ and −3.2‰ in $\delta^{13}C$ are documented[21]. Comparing them to the samples around the 550 ka lowstand (cores at 4.6 m and 5.6 m between 532 and 565 ka) which average $\delta^{18}O$ values of −16.4‰ and $\delta^{13}C$ of −1.7‰ suggests that glacial conditions prevailed around this time. Thus, it is concluded, that a climate-induced origin of this pronounced lowstand is very unlikely.

The global climate around this time is complex. The glacial MIS 14 was characterized by a smaller ice volume compared to MIS 12 or MIS 16[29]. The atmospheric $CO_2$ record shows a short (< 5 kyr) excursion to higher values around 557 ka while generally falling[6]. The Asian monsoon had short (< 5 kyr) oscillations between 550 ka and 560 ka[9]. Sea level was declining without any resolved short-term excursion[7]. Considering the known and described mechanisms for change in moisture availability in the Great Basin, there is no implication for enhanced local aridity during this time period.

In the absence of robust evidence for a climate-driven mechanism causing the 550 ka lowstand at Devils Hole, it is regarded as more likely that this was an event that affected the water table only locally, e.g., the opening of a (set of) tectonic fissure(s) that allowed for higher groundwater transmittance. Subsequently, this fissure would have had to be closed (e.g., by rock deformation or calcite deposition) to explain the return to a higher water table. If such changes had occurred in the fractured aquifer downstream of Devils Hole, they had no effect on the $\delta^{234}U_{initial}$ of the Devils Hole calcite and even if they occurred upstream, changes in $\delta^{234}U_{initial}$ would likely be small compared to the signal integrated over the 80 km long flow path from the source region to Devils Hole because Pérouse and Wernicke[28] show that no single outstandingly strong tectonic event took place.

**Long-term component of water-table decline**

Between 719 ± 14 ka and 438 ± 14 ka, there was an apparent long-term component of the decline in the water table over about 10 m, quantified by the gradual cessation of calcite deposition in the higher cores (dashed line in

Fig. 3). A sampling bias to explain this is considered unlikely because continuous deposition is documented for the last 750 kyr.

The end of the water-table decline coincides with the Mid-Brunhes event, a shift from less pronounced but longer warmth during interglacials before 430 ka to shorter and more pronounced ones afterward[30]. The water-table decline occurred over about 300 kyr and the Mid-Brunhes event was a change from one glacial cycle to another (< 100 kyr). Thus, the simultaneity of the end of the water-table decline and the Mid-Brunhes event is considered a coincidence. Yet, an amplification or dampening of the water-table change associated with the Mid-Brunhes event seems possible.

Condensation corrosion locally removed calcite from cave walls previously exposed to moist air. This effect has been observed in parts of the upper chamber in Devils Hole #2 and in a nearby cave[31]. Condensation corrosion starts a few meters above the water table and increases in intensity with height[31]. Corrosion increasing in intensity with elevation could lead to a gradual loss of calcite not unlike the gradual cessation of calcite layers with elevation (dashed line in Fig. 3), because it removes progressively younger calcite from cave walls with increasing elevation. However, folia and thin mammillary calcite layers younger than 438 ± 14 ka exist at +9.9 m elevation[11], while thick layers in the older half of the core indicate that highstands were longer and more frequent in the older part of the record. This shows that the cave wall at +9.9 m elevation previously had longer periods of submergence. The same observation can be made at all elevations, which qualitatively suggests a long-term component of decline of the mean water table. The rate of 3.4 m per 100 kyr is associated with a large systematic uncertainty, because no upper limit for the highstands is available for Devils Hole.

The origin of a non-climatically induced long-term decline of the groundwater table is the subject of debate. The Ash Meadows aquifer is located in an active extensional tectonic setting with fault slip rates of about 0.03 mm/yr over the last 500 kyr[28]. Perouse and Wernicke[28] proposed that normal fault slip rates in the Great Basin alternated in local clusters between an active mode lasting for about 50 kyr when slip rates are an order of magnitude higher than in slow periods lasting between 200 and 400 kyr. Fault movements in the aquifer do not directly affect groundwater levels at Devils Hole, but tectonic activity can cause changes in transmissivity at different places in the aquifer, which in turn can cause water-table changes. If tectonic activity was responsible for the long-term component of water-table decline at Devils Hole, then the proposed alternation between slow and fast periods is consistent with our data showing a negligible long-term component since 438 ± 14 ka and a higher rate before that.

The extension of the water-table record back to 750 ka allows the climate-driven fluctuations to be disentangled from the long-term tectonic component. Our record shows evidence of water-table declines and rises superimposed on a long-term declining trend between 719 ± 14 ka and 438 ± 14 ka of 3–4 m per 100 kyr. There is no apparent long-term component since 438 ± 14 ka (cf. also ref. 11). The rate of the long-term decline is

smaller than that suggested by Winograd and Szabo[22], because about half of the 26 m decline after 750 ka was caused by climate-induced changes because the vein seems to have been deposited during a glacial highstand according to our findings. The model of a slowed decline[22] is consistent with this study and ref. 11.

### Impact of terminations on the Devils Hole water table

Nine glacial-interglacial cycles with corresponding Terminations occurred during the last 820 kyr[32]. The global impact of Terminations is apparent in sea-level rises[7], shifts to higher atmospheric $CO_2$ levels[6] and, in the early phase of each Termination, intervals of weak Asian monsoon activity[9]. Wendt et al.[11] dated the youngest four Terminations at Devils Hole, and our study provides data back to Termination IX.

The Devils Hole data document a major decline in water table between $438 \pm 14$ ka and $403 \pm 6$ ka from more than $+9.9$ m to less than $+3.2$ m as well as a minimum in $\delta^{234}U_{initial}$ as expected from a major groundwater lowstand[25]. This timing is consistent with major rises in atmospheric $CO_2$[6] and global sea level[7] associated with Termination V (Fig. 5A, B). The Asian monsoon was weak between $430.5 \pm 1.5$ ka and $426 \pm 2$ ka, which agrees with the early phase of water-table decline at Devils Hole (Fig. 5D).

The water-table lowstand around 550 ka ($\pm 15$ ka) predates Termination VI compared to other climate archives[6,9,32]. We attribute this lowstand to local (tectonic) processes that were not linked to changes in recharge. The corresponding weak Asian monsoon interval lasted about 4.5 kyr and was centered at $532.3 \pm 3.5$ ka[9] which significantly predates the Devils Hole water-table decline at $515 \pm 10$ ka (mean of samples H374 and H380) recorded only in the core at $+9.9$ m. Considering that the water-table decline associated with Terminations and subsequent interglacials usually extends over ~20–30 kyr[11], the decline at $515 \pm 10$ ka may reflect the end of what would have been a longer decline period without the 550 ka non-climatic event. Continuous calcite deposition throughout this time at $+5.6$ m elevation indicates an interglacial lowstand above this elevation, higher than for any younger Termination. The magnitude of hydroclimate change in the Great Basin associated with this Termination can thus be considered as rather low. Similarly, a low magnitude of change at Termination VI is also observed in sea level[7] (Fig. 5B), atmospheric $CO_2$[6] (Fig. 5A) and Asian monsoon intensity[9] (Fig. 5D).

Termination VII is reflected in a water-table decline between $660 \pm 15$ ka and $627 \pm 15$ ka from more than 18.7 m to just below 6.8 m. Although small compared to their uncertainty, $\delta^{234}U_{initial}$ values also show a minimum at this time (Fig. 4), adding further evidence that this lowstand affected the whole aquifer. The timing is shifted towards older ages compared to the global sea level but still within uncertainty and is in good agreement with the Antarctic $CO_2$ record (Fig. 5). The weak Asian monsoon interval ended at $627 \pm 6$ ka[9] which coincides with the end of water-table decline in Devils Hole. Another weak Asian monsoon interval around 585 ka is associated with Termination VIIa. Our record shows a corresponding decrease at $573 \pm 14$ ka from above 13.3 m to below 6.8 m at $584 \pm 14$ ka (within uncertainty this is forward in time).

Termination VIII is reflected by a water-table decline after $719 \pm 14$ ka at 19.5 m, after $724 \pm 15$ ka at 18.7 m, and by a lowstand at 15.8 m after $720 \pm 14$ ka. All other cores show a 3 mm-thin layer of proto-folia (Supplementary Material Fig. S4), which was dated to between $722 \pm 16$ ka and $690 \pm 16$ ka at 6.8 m. Hence the water-table decline may have had an amplitude of more than 10 m, but did not last long enough below 15.8 m to form a thicker folia or proto-folia layer. $\delta^{234}U_{initial}$ values show a minimum comparable to their uncertainty (Fig. 4) indicating a lowstand, although with low certainty.

Termination IX falls into the gap of the earliest deposition at 820 ka and the onset of the continuous water-table record around 750 ka. The $\delta^{234}U_{initial}$ values also cannot resolve an increasing trend after this Termination which would correspond to a lowstand of high magnitude and long duration due to the high uncertainties.

Although uncertainties of 60.5‰ for $\delta^{234}U_{initial}$ are about one third of the total observed variability (between 1650 and 1850‰) and considering their indirect measurement via stable oxygen and carbon isotopes, it is worth noting that Terminations before the Mid-Brunhes event (i.e., Termination VI and older) were not accompanied by a shift in $\delta^{234}U_{initial}$ over 200‰, as seen for Terminations II, IV[25], and V. A smaller shift in $\delta^{234}U_{initial}$ could imply that the interglacials following these Terminations were not as arid or were relatively shorter in duration, or a combination of both.

### Insolation-induced ice-volume changes in North America as the driver of regional moisture availability during moderately warm climate states

The Devils Hole water table exhibited multiple higher-frequency (20–30 kyr) oscillations during three time periods: 75–110 ka (MIS 5 a–d), 470–530 ka (MIS 13), and 570–620 (MIS 15). These periods are moderately warm climate intervals characterized by intermediate CO2 concentrations[6] and global sea-level variations between −10 and −70 m relative to today[7]. Under these moderate climate conditions, Devils Hole water-table fluctuations show a close alignment with 65°N mid-July insolation (Fig. 5C) which has a cyclicity mode of about 22 kyr[33].

The periodicity of Devils Hole water-table fluctuations during MIS 5, 13, and 15 points to an indirect link to the Northern Hemisphere high latitudes. Moisture delivery to the Great Basin is largely facilitated by midlatitude westerly storms. The expansion of North American ice sheets is associated with a southward shift of the westerly storm track and subsequent wetting of the Great Basin[34,35]. Although the detailed mechanisms are debated, global climate model simulations suggest that North American ice sheets modulate the strength and location of semi-permanent pressure systems over the eastern North Pacific and North American continent, which steer westerly storm trajectories[15,16,36,37]. Insolation-forced changes in Arctic Sea ice extent and snow cover may have evoked similar atmospheric changes[38], as suggested by ~20 kyr hydroclimate swings in the Great Basin hydroclimate during MIS 5 a–d[34] when North American ice sheets were intermediate in size.

The extent of North American ice and snow cover during moderately warm climates remains poorly constrained. However, coupled models suggest that the alpine-based Cordilleran ice sheet was highly sensitive to summer insolation changes under intermediate interglacial conditions, such as MIS 13[39]. Insolation-forced expansions and retreats of intermediate-sized ice sheets coupled with variations in Arctic sea ice and snow cover may have influenced the westerly storm track position. Atmosphere-ocean feedbacks[40] and the impact of high-latitude cooling on the North American monsoon[18] may have also played a role. Overall, our results highlight a potential link between the Great Basin hydroclimate and insolation-forced changes in the cryosphere during intermediate warm periods.

## Methods
### Study area description

The Ash Meadows Groundwater Basin (11,500 km²; Fig. 1) has an annual recharge of 26,000,000 m³ (see ref. 17) and is part of the southern Great Basin (western USA). Today 80% of the basin's recharge originates from snowmelt in the Spring Mountains and Sheep Range[17,41]. The Ash Meadows Groundwater Basin has one major discharge zone in its southwestern corner that supports a desert oasis with endemic and endangered species, including the Devils Hole pupfish (*Cyprinodon diabolis*). The fish live in Devils Hole proper while the calcite samples of this study originate from Devils Hole #2, a second subvertical fracture intersecting the groundwater table 200 m to the north of Devils Hole proper (36.416°N, 116.283°W; Fig. 1C). The groundwater-table elevation is the same in both caves (based on our own survey), suggesting that they are hydraulically connected. For simplicity, both caves are referred to as "Devils Hole" in this study.

Calcite has been depositing from the slightly supersaturated groundwater onto the cave walls of Devils Hole[42]. Remarkably, despite variable water-table levels and chemical composition of the water, the temperature of the slightly thermal groundwater at Devils Hole has remained constant at 33–34 °C over the last 570 kyr[43,44]. Devils Hole calcite displays two distinct

**Fig. 5 | Comparison of Devils Hole water table with other paleoclimate archives.** Comparison between the Devils Hole water-table evolution and **A** atmospheric $CO_2$ from Antarctic ice cores[6], **B** global sea level[7], **C** mid-July 65°N insolation[33], and **D** the Asian monsoon[9]. Roman numbers refer to Terminations[32] as seen in major shifts towards higher $CO_2$ values or higher sea levels. Blue shading and gray line in the background is the interpreted qualitative probability of water-table height between 750 ka and 0 ka based on the data shown in Fig. 3. Note the inverted y-axes to better recognize the trends.

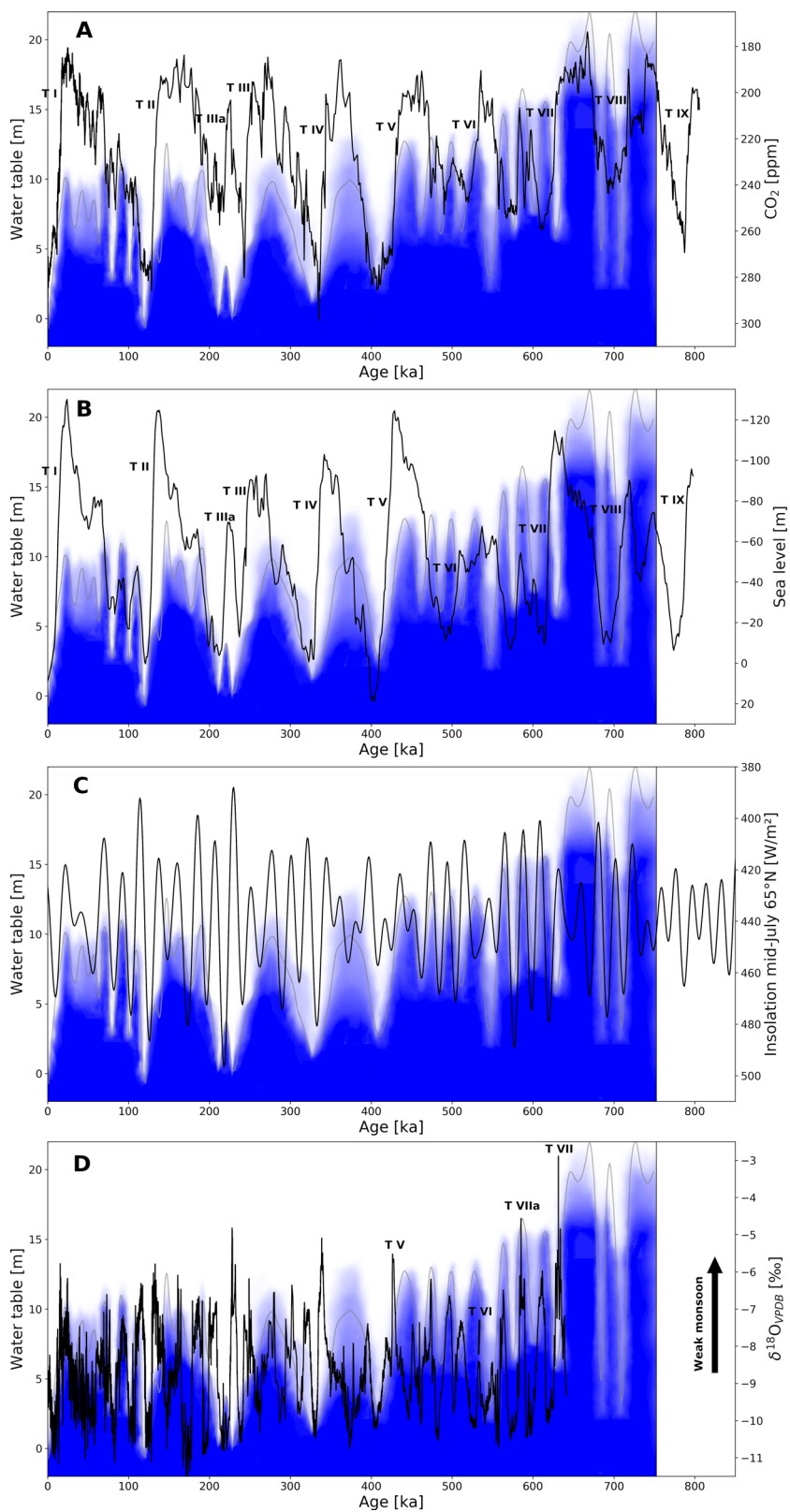

fabrics depending on the location of deposition with respect to the water table. In the first instance, cave walls that were continuously submerged are coated by slow-growing ($0.9 \pm 0.3$ mm/kyr[26]), dense mammillary calcite that can be reliably radiometrically dated using standard $^{230}$Th–U techniques[21]. The second type of fabric, known as folia, is deposited at the water table as a white, porous, faster-growing calcite that forms exclusively on the hanging wall of the cave (unlike the mammillary calcite that forms on both the footwall and hanging wall). Folia shows an open system behavior with respect to uranium isotopes and therefore cannot be dated reliably using the $^{230}$Th–U technique. Gradual transitions occur in the record between mammillary and folia calcite (see Supplementary Material for additional information).

A significant (about half a meter or more) rise of the water table causes folia deposits to become permanently submerged and overgrown by mammillary calcite. Dating the first mammillary calcite deposited on top of folia constrains the age of a water-table rise at the respective elevation. Conversely, a drop of the water table may cause folia to form on a previously submerged mammillary calcite-coated wall segment. Dating the uppermost mammillary calcite underneath such folia constrains the age of this water-table decline at the respective elevation[11].

## Analytical methods

Eleven cores of 2.5 cm diameter and 0.1–1.3 m length were drilled horizontally into the calcite deposits of the hanging wall of Devils Hole #2 at elevations between +3.2 m and +19.5 m relative to the modern water table. The cores were cut in half and polished. Calcite that appears macroscopically white was associated with folia (see Supplementary material section "Types of calcite deposits and their relationship to the water table" and Supplementary Table S1 for additional information on folia and its subtype referred to as proto-folia). In each core, samples of mammillary calcite adjacent to folia were sampled for U-series dating. Mammillary calcite samples taken immediately (<5 mm) underneath a folia layer are categorized to mark the decrease in the water table. Conversely, mammillary calcite samples just above (< 5 mm) a folia layer are categorized to record an increase in the water table. Samples of thin (< 10 mm) mammillary calcite layers bracketed by folia (i.e., <5 mm above prior folia and <5 mm below subsequent folia) are attributed to brief rises of the water table. Mammillary calcite samples more than 5 mm from any folia are categorized to mark submergence. This approach is similar to the one used by Wendt et al.[11]. Samples for dating (20–50 mg) were drilled perpendicular to the growth axis (Supplementary Figs. S1 and S2), and two aliquots (0.2–0.3 mg each) were used for stable isotope analyses.

U-series dating was performed at the University of Minnesota (USA). Samples were digested in $HNO_3$ and spiked with a mixed $^{233}U$-$^{236}U$-$^{229}Th$ spike similar to that described in ref. 45. Spiked samples were co-precipitated with Fe, centrifuged, and loaded into anion exchange columns following the methods described in refs. 46,47. Separate uranium and thorium liquid extracts were measured on a multi-collector inductively coupled plasma mass spectrometer (Thermo Neptune Plus) via a secondary electron multiplier using a peak-jumping mode[8,47]. Ages were calculated using the $^{230}Th$ and $^{234}U$ half-lives of ref. 8. Chemical blanks were measured with each set of 12 samples and were found to be negligible (< 50 ag $^{230}Th$, <150 ag $^{234}U$, <150 fg $^{232}Th$, <1 pg $^{238}U$). In order to derive $^{234}U$–U ages, $\delta^{234}U_{initial}$ was calculated from stable oxygen and carbon isotopes using the regression equations of Li et al.[26]. Its exponential decay to $\delta^{234}U_{measured}$ constrains the age. All uncertainties are given as 2σ ranges.

Oxygen and carbon isotope analyses were performed at the University of Innsbruck. The sample powders were analyzed using a semi-automated device (Gasbench II) linked to an isotope ratio mass spectrometer (ThermoFisher Delta V). Isotope values are reported relative to VPDB. Long-term precision is better than 0.1‰ for both $\delta^{13}C$ and $\delta^{18}O$[48]. The carbon and oxygen values of the duplicates from each sample replicated typically within ±0.1‰, never exceeding 0.2 ‰, and the mean value was taken.

## Data availability
All data are released with this study as supplemental material.

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

## Acknowledgements

This research was funded in whole or in part by the Austrian Science Fund (FWF) grants P263050 and P327510. For open access purposes, the author has applied a CC BY public copyright license to any author accepted manuscript version arising from this submission. Fieldwork was conducted under the scientific research and collecting permits of the United States National Park Services DEVA-2022-SCI-0014, DEVA-2017-SCI-0002 and DEVA-2015-SCI-0006. K.W. thanks the continued support of the Heising-Simons Foundation. This work was also partially supported by NSF Grant 2202913 to RLE.

## Author contributions

S.D.S. contributed to the design of the study, investigations, formal analysis, and writing the original draft. K.W. contributed to the conceptualization, investigations, and review of the draft. Y.D. contributed with investigations, supervision and reviewing the draft. R.L.E. contributed to infrastructure and reviewing the draft. X.L. contributed with formal analysis and reviewing the draft. G.M.C. contributed with formal analysis. G.E.M. contributed to the conceptualization and reviewing of the draft. C.S. contributed with conceptualization, investigations, funding acquisition, supervision, infrastructure and reviewing the draft. All authors contributed to the final version of the manuscript.

## Competing interests
The authors declare no competing interests.
