## [Transparent Peer Review file · Communications Earth & Environment]

Web links to the author's journal account have been redacted from the decision letters as indicated to maintain confidentiality.

Decision letter and referee reports: first round

30th Jan 24

Dear Mr Steidle,

I hope this email finds you well. I apologize for the delay in sending you our editorial decision.

Your manuscript titled "Moisture availability and groundwater recharge paced by orbital forcing over the past 750,000 years in the southwestern USA" has now been seen by 3 reviewers, and we include their comments at the end of this message. They find your work of interest, but some important points are raised. We are interested in the possibility of publishing your study in Communications Earth & Environment, but would like to consider your responses to these concerns and assess a revised manuscript before we make a final decision on publication.

We therefore invite you to revise and resubmit your manuscript, along with a point-by-point response that takes into account the points raised. Please highlight all changes in the manuscript text file.

In particular, please ensure that the revised manuscript meets the following editorial thresholds:

*** Restructure your introduction, clearly providing your research hypothesis and expanding the conceptual framework of your research.

*** Provide full justification for your methods and assumptions, including for the delta 234U initial regression model, and discuss uncertainties and potential systematic errors of your methods.

*** Expand the description of your results, and clearly describe all the elements interpreted in the discussion section.

Please use the following link to submit your revised manuscript, point-by-point response to the referees' comments (which should be in a separate document to any cover letter), a tracked-changes version of the manuscript (as a PDF file) and the completed checklist:

[redacted]

We hope to receive your revised paper within six weeks; please let us know if you aren't able to submit it within this time so that we can discuss how best to proceed. If we don't hear from you, and the revision process takes significantly longer, we may close your file. In this event, we will still be happy to reconsider your paper at a later date, as long as nothing similar has been accepted for publication at Communications Earth & Environment or published elsewhere in the meantime.

Please do not hesitate to contact us if you have any questions or would like to discuss these revisions further. We look forward to seeing the revised manuscript and thank you for the opportunity to review

your work.

Best regards,

Carolina Ortiz Guerrero
Associate Editor
Communications Earth & Environment

EDITORIAL POLICIES AND FORMATTING

Editorial Policy: Policy requirements (Download the link to your computer as a PDF.)

Furthermore, please align your manuscript with our format requirements, which are summarized on the following checklist:

Communications Earth & Environment formatting checklist

and also in our style and formatting guide Communications Earth & Environment formatting guide .

*** DATA: Communications Earth & Environment endorses the principles of the Enabling FAIR data project (<http://www.copdess.org/enabling-fair-data-project/>). We ask authors to make the data that support their conclusions available in permanent, publically accessible data repositories. (Please contact the editor if you are unable to make your data available).

All Communications Earth & Environment manuscripts must include a section titled "Data Availability" at the end of the Methods section or main text (if no Methods). More information on this policy, is available at <http://www.nature.com/authors/policies/data/data-availability-statements-data-citations.pdf>.

If a community resource is unavailable, data can be submitted to generalist repositories such as figshare or Dryad Digital Repository. Please provide a unique identifier for the data (for example a DOI or a permanent URL) in the data availability statement, if possible. If the repository does not provide identifiers, we encourage authors to supply the search terms that will return the data. For data that have been obtained from publically available sources, please provide a URL and the specific data product name in the data availability statement. Data with a DOI should be further cited in the methods reference section.

REVIEWER COMMENTS:

Reviewer #1 (Remarks to the Author):

The Devils Hole record is a foundational record of western US hydroclimate, forming one of the longest continuous records that exists over the last 500 ka in this region. It has formed one of the canonical climate records with which other climate records from this region are compared and contrasted, akin to how the Hulu Cave record is utilized in Asian paleoclimate comparisons. Substantial work within the past decade has focused on bringing coherence to the geochronology at Devils Hole, and new and extended age models have been utilized to look in further detail at proxy records of rainfall, local water-table changes, and tectonics among other things. The current study employs predominantly ^{234}U -U dating using $\delta^{234}\text{U}$ initial estimates based on modeled regressions to $\delta^{18}\text{O}$ and $\delta^{13}\text{C}$ data, to extend the existing Devils Hole water table record from 350 to 750 ka ago. The authors interpret this older part of the water table record to be composed of long-term local tectonic forcing and broader-scale hydroclimate forcing. Due to the wide use of the Devils Hole data and its importance to both climate reconstruction comparisons and as a baseline for climate model efforts, this study extending the water table record back 100s of ka and utilizing a relatively new method of extending U-Th dating beyond traditional timescales (using the ^{234}U -U dating method, via modeled estimates of $\delta^{234}\text{U}$ initial) is of interest to readers of *Communications Earth & Environment*.

I found the paper overall to be well written and clearly structured. The inclusion of detailed supplementary data and figures was very helpful to assess the main arguments of the paper. However, I do think that the manuscript could benefit substantially in a few areas from some more rigorous data analysis, and clearer explanation of decisions used when plotting or talking about subsets of the data, and expansion of reasoning behind some key assumptions made in order to reach the stated conclusions. This is crucial as the water table record, and inferences drawn from it throughout the discussion, are underpinned by the data set itself. I recommend that a substantial revision is warranted to deal with the data and assumptions more robustly before this paper is ready for publication and explain comments in more detail below.

Major comments:

1. Robustness of assumptions on $\delta^{234}\text{U}$ initial regression model

The regression equations of Li et al. (2020) are used to translate $\delta^{18}\text{O}$ and $\delta^{13}\text{C}$ measurements into $\delta^{234}\text{U}$ initial values with which many of the ^{234}U -U ages are calculated. How robust is the assumption that the regression can be extended even further beyond the calibration dataset than the original publication? There is very little written in either Li et al., 2020 or in this manuscript to justify pushing this regression deeper in time. I would like the authors to expand on their reasoning that we can assume the modeled relationship between $\delta^{18}\text{O}$ and $\delta^{13}\text{C}$, and $\delta^{234}\text{U}$ initial, holds this far beyond the calibration dataset. In a similar vein and on a related topic, the manuscript also discusses that the conclusions of Wendt et al. (2020) on other processes that could affect $\delta^{234}\text{U}$ initial besides water table elevation can be extended back to 750 ka, with very little discussion or reasoning to justify this assumption. This should also be expanded upon. Unless thorough justification can be laid out to the contrary, it would seem that a 60 ‰ uncertainty on the $\delta^{234}\text{U}$ initial is a MINIMUM expected error as this modeled relationship is pushed further from the calibration dataset. This should be clearly stated as a minimum uncertainty estimate in the manuscript, and plotted as such on the figures.

2. Comparison of ^{230}Th -U ages with ^{234}U -U ages

a. Line 195: The term "1.5x their combined uncertainty" does not convey meaningful information when it comes to this dataset. Consider more robust description/quantification of closeness of fit (see below point).

b. Line 241: Employ a more robust treatment of the 1:1 plot, e.g., bootstrap the data and calculate a range of R² and p values. This will allow quantification of uncertainties on the R² and p values which can be used to assess the strength of the argument in age comparison. Please state the actual p-value (not just a < number), and state the null hypothesis used in calculating it (i.e., that the ages are not concordant).

3. Inclusion/exclusion of samples from the data set for use

a. Line 194: The samples being discussed here in terms of agreement between ²³⁰Th-U and ²³⁴U-U ages appears to be from the total of 92 samples, however it is mentioned in Line 177 that only 59 of the samples could have ages derived from both dating methods (based on set uncertainty criteria). This is confusing to the reader; please make it clear which data you are using and why.

b. Line 241: Why are only 53 of the 59 ²³⁰Th-U ages being used and 86 of the ²³⁴U-U ages? Why do the ²³⁰Th-U ages in Panel B plateau at 650 ka when we clearly see older samples in Panel A; if it is simply because they would plot off the chosen axis scale then this should be clearly stated or better show these with a broken scale bar in order to actually let the reader see all of the available data. Any exclusions, or decisions made on what data to plot or exclude, needs to be more clearly laid out.

Minor comments:

4. Line 15: The letter "c" is missing from the word "correspondence."

5. Line 41: Add space between number and % sign.

6. Line 42: Used UK spelling of "modelling"; if this is meant to be standardized to American use "modeling."

7. Line 129: Panel 1B - the blue arrows are showing up quite blurred and hard to tell which direction they point in; I suggest making these show up more sharply. Panel 1C - the location lines for Cores K and L have a brown colored strike line through for no apparent reason: either explain the reason in the figure caption or remove the strikes. Panel 1C - for the label for Core "Ibis", this is written as "I-bis" in the supplemental material: please align the naming between these files.

8. Line 96: Add space between number and % sign.

9. Line 102: Add space between number and % sign.

10. Line 131: There is a period missing at the end of "Overview map of North America."

11. Line 165: The comma is missing after "Vennemann" in the reference.

12. Lines 165-167: Add spaces between numbers and ‰ signs.

13. Line 178: There is an extra space between the "<" and "1."

14. Line 179: The language "agree within uncertainty" seems an overreach here. In examining Panel 2A there are many ages which appear to disagree outside of the plotted uncertainty. Softened language along the lines of "mostly agree within uncertainty" would seem more reasonable.

15. Line 182: There is an extra space between the ">" and "450."

16. Line 190: There is an extra space between the ">" and "9.91."

17. Lines 193-195: Add spaces between numbers and % or ‰ signs.

18. Line 209: Specify, perhaps in terms of relative %, more precisely what the "thicker" and "thinner" variations mentioned are, i.e., are mammillary calcite layers 50 % thinner in recent glacial periods?
19. Line 228: The comma is missing after "i.e."
20. Line 232: The unit ka is missing after "555±13."
21. Line 233: The space is missing between "+6.8" and "m."
22. Line 239: The comma is missing after "i.e."
23. Line 241: State whether age uncertainties plotted in Panel A are 1 or 2 sigma on key or in caption, and on the Panel B y-axis.
24. Line 243: Suggest editing "230Th-U and 234U-U age dates" to "230Th-U and 234U-U ages."
25. Line 244: "234U age" should be edited to "234U-U age."
26. Line 251: Suggest adding rate of decline to the dashed line. Some of the sample symbols are not as clear as the others, it is unclear why but perhaps they are not all on the same layer within the figure, please make sure all are easily legible (one glaring example is the sample at +3.2 m, 500 ka). May be useful to add a small visual key showing what the dense blue, white, and intermediate colors mean rather than stating it in caption; would make the figure easier to grasp at first glance. Are the black horizontal lines through the data symbols intended to show age uncertainty? If so, please state this in the figure caption, and whether this is a 1 or 2 sigma uncertainty.
27. Line 394: Remove the space between "±" and "15."
28. Line 417: The decrease in water-table is quoted as being from 573 to 584 ka, it seems like this should be inverted to move forward in time to present, and not backwards.
29. Line 432: The comma is missing after "i.e."
30. Line 466: It could be helpful to the reader to annotate Panel D with arrows indicating direction of strong/weak monsoon.
31. Line 467: It is noted that Dataset B is on an inverted Y axis, however Datasets A and C are both also on inverted Y axes. Potentially worth noting this too.
32. Line 469: Dataset D is described as "the Asian monsoon" which is inaccurate. Please adjust this to be an accurate description, like "δ18O record used as proxy for the Asian monsoon."

Supplemental Material:

33. Line 39: The main text uses the notation "Devils Hole #2" while the notation "Devils Hole II" is used here. Please align these.
34. Line 53: The main text uses the notation "Devils Hole #2" while the notation "Devils Hole II" is used here. Please align these.
35. Line 65: Change to "Figure S2" as Sample G-426 is shown in Figure S2, not Figure S1.
36. Line 69: Figure S1 - state that these images do not show the top of all cores, e.g., Cores H and I-

bis. Is there a reason for this, are these images published elsewhere? If so, it would be helpful to state the reason / reference relevant publication.

37. Line 72; Change "I:" to "I-bis:" as the core image in this figure is for Core I-bis.

38. Line 74: Figure S2 - state that these images do not show the top of any of the cores. Is there a reason for this, are these images published elsewhere? If so, it would be helpful to state the reason / reference relevant publication.

39. Line 98: Figure S5 - the left overview scale bar labeled "1 mm" does not make sense given the zoomed in microscope scale bar, I believe that the overview scale bar should be labeled "1 cm" so please check and adjust accordingly.

40. Line 112: The ":" is missing after the Figure S8 title.

41. Line 113: Capital letter missing from "figure 3." Perhaps the "Note" that is found at the bottom of Figure S9 would be useful to add to this figure caption too as the individual uncertainties are also difficult to read.

42. Line 115: The ":" is missing after the Figure S9 title. Capital letter missing from "figure 3."

43. Line 116: Capital letter missing from "figure S8."

U-Th Data Set:

44. Description Tab, Line 6: Correct the spelling of "sumbergence"; the b and m are inverted.

45. Description Tab, Line 18: Correct "V&X" to "W&X."

Reviewer #2 (Remarks to the Author):

I greatly enjoyed reading this manuscript, which builds on the earlier work of Wendt et al. (2018) and Szabo et al. (1988) to piece together the groundwater history of (part of) the Great Basin USA using Devils Hole vein calcites over the last 800-odd kyr. The data presented is very impressive indeed, and I cannot think of any places worldwide where such a detailed chronology of long-term aquifer recharge history has been assembled. Due to the chronology, the authors are also able to place their reconstruction in the context of orbital forcing and paleoclimate. They also provide new insights on the region's tectonic history. Therefore, this paper will attract wide interest in the Earth Science community and is potentially suitable for publication in COMMSENV.

The paper is well illustrated and the quality of the writing is outstanding (this makes a refreshing change!), with only a few minor edits marked on the Word docs. I also appreciated the level of detail provided in the Table S2. The supplementary information is very useful - it is great to see the cores and thin-section images, which will be appreciated by many in the speleothem community.

My main criticism relates to the narrative, or pitch, of the ms. This is immediately evident in the abstract, which, although very informative, neither identifies nor contextualises the wider scientific problem being addressed by the paper. This would seem out of alignment with the requirement for most Nature journals (including this one, I assume?). The issue somewhat continues into the Introduction, where there is background and rationale provided (as per the journal guidelines) but the background is almost exclusively related to previous studies at the site/in the region (which is necessary) with little by way of background or context to the wider scientific problem. For example, the content of the first part of paragraph #2 could be expanded to become the lead paragraph of the

Introduction section that exposes the wider/bigger issue being addressed, after which comes the brief site info/previous research and paves the way for this new work. I guess what I am suggesting is that perhaps the ms should be tweaked to make it more suitable to COMMSENV. As it stands, in spite of the excellent data/results, the ms appears more appropriate for EPSL or Geology (but it easily has the potential to be in this journal because of the quality and uniqueness of the data).

Reviewer #3 (Remarks to the Author):

The manuscript provides a new and extended Devil's Hole record covering over 750,000 years of groundwater variability. The authors conduct meticulous work on mammillary calcite and folia formations to derive robust ages to constrain groundwater variability well outside the limits of traditional U-Th ages, ~600,000 years. In their discussion, the authors try and associate groundwater variability within Devils Hole/ d234U to tectonic and climate processes. The manuscript will be valuable from a geochronological, paleoclimatological, paleohydrology, as well as tectonics perspective. The publication of the manuscript I believe will further encourage conversations and novel research between fields such as karst hydrology, speleothem sciences, paleoclimatology, and sedimentology, which is required to solve signals from this complex system. I enjoyed reading the manuscript though I think addressing certain statements and assumptions (see attached document) will further strengthen the paper.

I suggest minor revisions and upon revision would be supportive of this paper's publication.

1 SUPPLEMENTAL MATERIAL

**Types of calcite deposits and their relationship to the water table**

The water table in Devils Hole undergoes diurnal fluctuations on a scale of a few centimeters due
to Earth tides and weather events (Cuttillo and Ge, 2006). As a result of these fluctuations, the
water table oscillates within a few centimeters up and down the cave wall. This means that a
small range of the cave wall alternates between being submerged in water and being exposed to
air. On timescales of millennia, these small-scale oscillations are superimposed on larger-scale
rises and falls of the groundwater table and are relevant for the formation of folia.

**Mammillary calcite** forms below the minima of diurnal water table oscillations, i.e., it is a
subaqueous deposit. This speleothem type is translucent in thin section and has very few
impurities and inclusions compared to folia. Its fabric is compact, comprising composite
columnar crystals, each being a bundle of multiple rod-shaped crystallites (Figures S3-5). Fluid
inclusions are rare (Figures S3G and S6A) and occur locally at the boundaries of adjacent
bundles of crystallites (Figure S6A). Although mammillary calcite is known from other caves
(also referred to as “cave clouds” – e.g., Hill and Forti, 1997; Polyak et al., 2008), the
microscopic fabric described above seems to be unique to Devils Hole.

**Folia** occurs as mm to cm-thick and commonly porous layers in many cores. Folia is
conspicuously white in reflected light and consists of mosaic to elongated calcite crystals with
abundant fluid inclusions and pores ranging in size from μm to mm (Figure S5). While hiatuses
locally exist within folia, the boundary to the under- and overlying mammillary calcite is often
gradual (Figure S5; D’Angeli et al., 2015; Kolesar and Riggs, 2004).

Folia has been reported from several caves (e.g., D’Angeli et al., 2015; López Martínez et al.,
2015). Its formation is restricted to hanging cave walls and is associated with a fluctuating water
table (Kolesar and Riggs, 2004; Davis, 2012; D’Angeli et al., 2015).

**Proto-folia.** Thick and porous folia layers are increasingly rare in the older parts of the cores.
Yet mm-thin and macroscopically white layers are sometimes present between mammillary layers.
Petrographic examination of thin sections shows that the fabric of these white layers is identical
to that of the mammillary calcite but contain abundant, sometimes large fluid inclusions of
irregular shape (Figures S3, S4, S6B and C). Rarely, fluid inclusions contain small vapor bubbles
(Figure S6B), which may be explained by either entrapment of air when temporarily emerged
from water, or by entrapment of exsolved CO_2 bubbles. We refer to these layers as proto-folia,
and consider them as precursors of well-developed folia. We are not aware of reports of this type
of fabric in caves elsewhere.

The change from mammillary calcite to proto-folia is gradual and lacks sharp changes in the
crystal fabric (Figure S3). Within a few mm, these white layers typically become more porous
(Figure S3) after which a transition back to mammillary calcite occurs (Figures S1 and S2).

Cores more than 10 m above the modern water table were drilled in the near-vertical non-
overhanging wall in Devils Hole II, where the accretion of adherent particles was suppressed and
therefore folia did not form. This agrees with observations in Devils Hole proper, where folia is
only present where the wall is slightly overhanging (Kolesar and Riggs, 2004). Frequent
alternations between submergence and exposure to air due to diurnal water table oscillations

likely led to enhanced CO₂ degassing and non-equilibrium calcite precipitation conditions at wall
 segments located at the water/air interface, giving rise to a high abundance of aqueous inclusions
 and some porosity.

The abundance of proto-folia in older cores compared to proper folia in younger cores could also
 be related to a change of the cave microclimate associated with the gradual opening of the cave
 to the surface by erosion, and/or the availability of detrital particles. In summary, we interpret the
 proto-folia layers to reflect deposition close to the water table and thus use them as indicators of
 rises or falls of the paleo-water table the same way folia is used. Table S1 summarizes the main
 characteristics of mammillary calcite, folia and proto-folia.

Table S1: Petrographic characteristics of the three speleothem types present in Devils Hole II and
 Devils Hole proper.

	Mammillary calcite	Proto-folia	Folia
Color (macroscopic)	Translucent	White and opaque	White and opaque
Macroscopic growth structure	Massive and uniformly thick coating of coarsely crystalline calcite on most underwater surfaces in Devils Hole. The outer surface forms dome-shaped protuberances (Kolesar and Riggs, 2004)	M mm-thin layers lacking distinct macroscopic growth structures. Only found in cores, not present at today's cave wall	Wedge-shaped in vertical section (Kolesar and Riggs, 2004)
Fabric	Compact fabric of composite columnar crystals, each being a bundle of multiple rod-shaped crystallites	Same fabric as mammillary calcite but containing much more inclusions	Consists of small (a few tenths of a mm to a few mm) equant to slightly elongate length-fast calcite crystals. Initial precipitates exhibit a dendritic habit, with most of the growth occurring on the upper surface of folia. With continued calcite precipitation, up to 10 mm-long columnar, length-fast calcite crystals may occur (Kolesar and Riggs, 2004)
Boundaries	None	Gradual changes	Sharp boundaries between different types of fabrics exist but also gradual changes

Inclusions and pores	Dense calcite with less than 1% porosity. Pore spaces are of two different types: (a) irregularly shaped pore spaces with fluid inclusions that are commonly oriented parallel with the crystallite boundaries. (b) Pore spaces formed by debris settling onto up-facing mammillary calcite surfaces in the plane of the crystal terminations, i.e., parallel to the growth surface (Kolesar and Riggs, 2004)	Variously abundant fluid inclusions	Abundant fluid inclusions and open pores
Formation	Subaqueously, below the range of diurnal water table oscillations	Mostly subaqueously but very close to the water table where enhanced degassing of CO ₂ is expected to occur. Occasional exposure to air.	Within the range of water table oscillations

Samples out of stratigraphic order

Four (N-754, N-749, J-813 and F-573) out of the five omitted samples were discarded because they are out of stratigraphic order. They are from thin layers bracketed by petrographic boundaries or rest directly on bedrock. These samples are interpreted as younger deposits whose origin is possibly related to small fractures in the calcite deposit which were later filled by calcite during a subsequent highstand. Sample G-426 was treated as an outlier and omitted because it is too old with respect to the 2-sigma uncertainty compared to G-425 and G-497 (Figure S1).

Figure S1: Cores of Devils Hole calcite drilled horizontally into the cave wall (see Figure 1). Folia and proto-folia deposits appear white while mammillary calcite appears grey and at few places brown. Bedrock and rock fragments are annotated with a "B". Black lines are drilling positions for dating of mammillary calcite (letter = name of the core; number = mm from the top of the core). Elevation above present day water table of different cores: M: 18.7m; R: 19.5m; L: 15.8m; K: 13.3m; H: 9.9m; I: 8.3m.

Figure S2: Cores of Devils Hole calcite (continued from Figure S1) drilled horizontally into the cave wall (see Figure 1). Folia and proto-folia deposits appear white while mammillary calcite appears grey and at few places brown. Bedrock and rock fragments are annotated with a "B". Black lines are drilling positions for dating in mammillary calcite (letter = name of the core; number = mm from the top of the core). Elevation above present day water table of different cores: I: 8.3m; N: 6.8m; J: 5.6m; G: 4.6m; F: 3.2m.

Figure S3: Transition from mammillary calcite to proto-folia. A: Overview of the middle part of drill core G (see Figure S2). Plane-polarized (B) and cross-polarized (C) transmitted-light photomicrographs showing the gradual transition from mammillary calcite to proto-folia. D-G: Scanning electron microscope images of the surface of a polished thin section showing proto-folia (D-F) and mammillary calcite (G). D – 6 mm from the mammillary calcite layer; E – 3 mm from the mammillary calcite layer; F – 1 mm from the mammillary calcite layer; G – within the mammillary calcite layer. Some pores associated with fluid inclusions (FI) and interconnected pores (IP) are marked.

Figure S4: Two thin white proto-folia layers bracketed by mammillary calcite. A: Overview of the lower part of drill core I-bis (see Figure S1). Plane-polarized (B&D) and cross-polarized (C&E) transmitted-light photomicrographs of the younger (B&C) and older (D&E) proto-folia layer.

Figure S5: Transition from folia to mammillary calcite.. Left: Overview of the middle part of drill core H (see Figure S1). Right: Picture of a thin section under cross-polarized transmitted light.

*Figure S6: Fluid inclusions (FI) in mammillary calcite (A) and proto-foia (B and C) under plain-polarized light. A – three single-*
 *phase (all-liquid) inclusions alligned along the compromise boundary between adjacent crystallites (dashed line); B – relatively*
 *large two-phase (liquid-vapor) inclusion; vapor bubble was either accidentally trapped air or represents exsolved CO₂; C -*
 *relatively large single-phase (all liquid) inclusion. Many other inclusions in B and C are out of focus.*

Figure S7: Visual aid of water table history (see Figure 3 and main text) used as template.

Figure S8 Spline function of the most likely water table history based on the presented water table markers. The data is attached as a supplement to the manuscript. Blue shading and water table markers are the same as in figure 3 in the main text.

Figure S9 Water table markers and blue shading as in figure 3 of the main text but zoomed in. Additionally, the spline function of the most likely water table history as in figure S8. Note: Uncertainties of individual data points older than 450 ka are difficult to read, but they are all in a narrow range between ± 14 ka and ± 17 ka.

REFERENCES

Cutillo and Ge, 2006 Analysis of strain-induced ground-water fluctuations at Devils Hole, Nevada
<https://doi.org/10.1111/j.1468-8123.2006.00150.x>

D'Angeli et al., 2015: Genesis of folia in a non-thermal epigenic cave (Matanzas, Cuba)
<http://dx.doi.org/10.1016/j.geomorph.2014.09.006>
Davis 2012: In defense of a fluctuating-interface, particle-accretion origin of folia
<http://dx.doi.org/10.5038/1827-806X.41.2.6>
Hill, C., Forti, P., 1997. Cave Minerals of the World. 2nd ed., Huntsville (National Speleological
Society).
Polyak, V., Hill, C., Asmerom, Y., 2008. Age and evolution of the Grand Canyon revealed by U-
Pb dating of water table-type speleothems. Science, 319, 1377-1380.
Kolesar, P. T. and A. C. Riggs, "Influence of depositional environment on Devils Hole calcite
morphology and petrology," in Studies of Cave Sediments, Ed.I. D. Sasowsky and J.
Mylroie Springer US, 2004, pp. 227–241.
López Martínez et al, 2020: Bubble trail and folia in cenote Zapote, Mexico: petrographic
evidence for abiotic precipitation driven by CO2 degassing below the water table
<https://doi.org/10.5038/1827-806X.49.3.2344>

Moisture availability and groundwater recharge paced by orbital forcing over the past 750,000 years in the southwestern USA

Simon D. Steidle^{1,c}, Kathleen A. Wendt², Yuri Dublyansky¹, R. Lawrence Edwards³, Xianglei Li^{3,4}, Gracelyn McClure³, Gina E. Moseley¹, Christoph Spötl¹

¹Institute of Geology, University of Innsbruck, Innrain 52, 6020 Innsbruck, Austria

²College of Earth, Ocean, and Atmospheric Sciences, Oregon State University, 101 SW 26th Street, Corvallis, Oregon 97330

³School of Earth and Environmental Sciences, University of Minnesota, 116 Church Street SE, Minneapolis, MN 55455-0149, USA

⁴Institute of Vertebrate Paleontology and Paleoanthropology, Chinese Academy of Science, 142 Xizhimenwai Street, Beijing 100044, China

^ccorrespondence: simon.steidle@student.uibk.ac.at

ABSTRACT (<150 words) (currently 149 words)

Calcite deposits in Devils Hole, Nevada, provide a unique record of regional groundwater-table fluctuations in the Great Basin. Extending this record back to 750,000 years documents multi-meter oscillations that closely align with regional and global climate transitions from humid glacials to arid interglacials. The timing of Terminations, marked by first-order drops in groundwater levels, exhibits a strong correspondence with global climate shifts, although with smaller magnitudes prior to 450,000 years. During periods of moderately warm climate, the water table displays a remarkable sensitivity to variations in Northern Hemisphere summer insolation. This connection may have transpired through the changing extent of North American ice sheets, which induced a southward shift of moisture-laden trajectories towards the southwestern USA. These orbitally driven hydroclimatic changes are superimposed on a long-term decline in the regional water table observed in the older half of the record, a trend that was likely influenced by tectonic factors.

INTRODUCTION

During the Quaternary, the southwestern region of North America repeatedly switched between arid, warm interglacial climates, as characterizes the region today, and humid, cooler glacial climates that led ~~regionally~~ to the filling of extensive lakes and higher groundwater tables (e.g., Reheis et al., 2019; Wendt et al., 2018; Santi et al., 2020; Seltzer et al., 2019). Increased moisture availability was closely tied to the expansion of the Laurentide ice sheet (Conroy et al., 2019; Oster et al., 2015) due to the presence of anticyclones that played a pivotal role in redirecting winter moisture from the northern Pacific further south into the Great Basin (Oster et al., 2015; Lora et al., 2017). Under contemporary conditions, winter precipitation remains the primary source of regional moisture, with the summer monsoon contributing only a minor portion (<10%) to annual local recharge (Halford and Jackson, 2020). Modelling results show that

Formatted: Strikethrough

during glacial periods, the summer monsoon was even further suppressed (Bhattacharya et al.,
2017).

Anthropogenically-forced climate change poses a significant threat to groundwater resources
and exacerbates socio-environmental conflicts in the southwestern USA (Deacon et al., 1991;
Deacon et al., 2007; MacDonald and Water, 2010). Reliable, independently-dated climate
archives that extend over multiple glacial cycles provide important quantitative constraints on
water availability under different climate regimes, thus enabling a deeper understanding of the
associated driving mechanisms and teleconnections. To enhance our comprehension of the
connection between global climate change and the mechanisms influencing regional
hydroclimate responses in the southwestern USA, we investigated the long-term variability of
moisture availability and groundwater recharge at Devils Hole, Nevada, spanning eight glacial-
interglacial cycles ~~and covering since~~ 750 ka (ka stands for thousand years ago).

Devils Hole, situated in the Ash Meadows Groundwater Basin of southern Nevada (Fig. 1), is
strategically positioned within the enclosed Great Basin. ~~Within~~ This subvertical extensional
fracture, intersecting the groundwater table, enabling large-scale atmospheric changes ~~are to be~~
preserved in submerged calcite deposits. These deposits, precipitated from groundwater onto the
cave walls, ~~enabled the reconstruction of the~~ have been shown to preserve a water-table elevation
history ~~over covering~~ the past 350 kyra (Szabo et al., 1994; Wendt et al, 2018; Moseley et al.,
2016). However, these calcite deposits extend much further back in time (up to 820 ka),
providing a unique opportunity to examine changes across several glacial-interglacial cycles and
to establish independent time constraints for understanding hydroclimate change in the Great
Basin.

The pioneering study of Winograd and Szabo (1988) reported calcite veins, dated to 750±50 ka,
perched at +26 m above the present-day water table on the surface of the Devils Hole ridge,
which were dated to 750±50 ka. Using this and two younger samples, these authors deduced a
mean rate of water table decline of 2-3 m/100 kyra over the past 750 kyra. They also noted a
more rapid decline of about 8 m/100 kyra around 700 ka ago, followed by a slower decline of 2
70 m/100 kyra over the last 510 kyra. Tectonic uplift was proposed as the primary driver for these
71 long-term trends (Winograd and Szabo, 1988). The link between water table changes and climate
emerged later (Szabo et al., 1994; Wendt et al., 2018), when climate-induced water-table
oscillations exceeding 10 m were reported, albeit without detecting a long-term trend over the
last 350 kyra. By extending the detailed history of water-table changes back to 750 ka, we
reconcile this discrepancy and quantify the relative contributions of climate-induced changes and
of the long-term component during the mid to late Quaternary.

The large water-table fluctuations over glacial-interglacial cycles at Devils Hole are also
paralleled by distinctive geochemical characteristics of the calcite (e.g., Winograd et al., 1988,
1992; Moseley et al., 2016; Wendt et al, 2020). Importantly, the relative abundance of uranium
isotopes in the calcite offers a qualitative link to the aquifer's water-table changes (Wendt et al.,
2020). A recent modelling study has quantitatively linked these changes to corresponding
changes in recharge of the Ash Meadows Groundwater Basin over the last 350 kyra (Jackson et
al., 2023). Remarkably, despite this variability in water-table levels and chemical composition
of the water, the temperature of the slightly thermal groundwater at Devils Hole has remained
constant at 33-34°C over the last 570 kyra (Kluge et al., 2014; Bajnai et al., 2021).

Commented [A1]: 5 ka = 5,000 years ago (wrt a to time reference, e.g. BP1950);
5 kyr = 5,000 years (a time span)

Calcite deposits found in caves are commonly dated using the ^{230}Th -U method, where the
radioactive decay of ^{238}U and ^{234}U results in the formation of insoluble ^{230}Th . Through an
iterative process, an initial ratio of $^{234}\text{U}/^{238}\text{U}$ ($\delta^{234}\text{U}_{\text{initial}}$) is estimated and the age (^{230}Th -U age) of
formation is calculated (Kaufman and Broecker, 1965). However, the short-lived ^{230}Th isotope
limits the applicability of this method to samples younger than about 650 ka (Cheng et al., 2013).
At Devils Hole, the age of ~~the~~ calcite deposits extends beyond this limit, offering an exceptional
opportunity to investigate a continuous terrestrial paleoclimate record covering eight glacial-
interglacial cycles. Establishing a reliable chronology beyond 650 ka is possible due to the
relatively stable $\delta^{234}\text{U}_{\text{initial}}$ values, which are far from secular equilibrium (1650 – 1850 ‰).
Furthermore, this dating method can be further corroborated by statistically significant
correlations with $\delta^{18}\text{O}$ and $\delta^{13}\text{C}$ derived from the same calcite samples (Li et al., 2020).
Consequently, the ^{234}U -U method proves to be a viable and dependable dating technique,
characterized by excellent precision (relative uncertainty of 2-3%), particularly for the older
parts of the Devils Hole record.

STUDY AREA DESCRIPTION

The Ash Meadows Groundwater Basin (11,500 km²; Figure 1) has an annual recharge of
26,000,000 m³ (Halford and Jackson, 2020) and is part of the southern Great Basin (western
USA). Today 80% of the ~~basin~~ recharge ~~to this basin~~ originates from snowmelt in the Spring
Mountains and Sheep Range (Winograd et al., 1998, Halford and Jackson, 2020). The Ash
Meadows Groundwater Basin has one major discharge zone in its south-western corner that
supports a desert oasis with endemic and endangered species, including the Devils Hole pupfish
(*Cyprinodon diabolis*). The fish live in Devils Hole proper while the calcite samples of this study
originate from Devils Hole #2, a second sub-vertical fracture intersecting the groundwater table
200 m to the north of Devils Hole proper (36.416°N, 116.283°W; Figure 1C). The groundwater
table ~~elevation shows is~~ the same ~~elevation~~ in both caves, suggesting that they are hydraulically
connected. For simplicity, both caves are referred to as “Devils Hole” in this study.

Calcite has been depositing from the slightly supersaturated groundwater onto the cave walls of
Devils Hole (Plummer et al., 2000). The calcite displays two distinct fabrics depending on the
location of deposition with respect to the water table. In the first instance, cave walls that were
continuously submerged are coated by slow growing (0.9±0.3 mm/~~keyr~~; Li et al., 2020), dense
mammillary calcite that can be reliably radiometrically dated using standard ^{230}Th -U techniques
(Moseley et al., 2016). The second type of fabric, known as folia, ~~is~~ deposited at the water table
as a white, porous, faster growing calcite that forms exclusively on the hanging wall of the cave
(unlike the mammillary calcite that forms on both the footwall and hanging wall). Folia has an
open system behavior with respect to uranium isotopes and cannot be ~~reliably~~-dated ~~reliably~~
using the ^{230}Th -U technique. Gradual transitions occur in the record between mammillary and
folia calcite (see Supplementary Material for additional information).

A significant (about half a meter or more) rise of the water table causes folia deposits to become
permanently submerged and overgrown by mammillary calcite. Dating the first mammillary
calcite deposited on top of folia constrains the age of a water table rise at the respective
elevation. Conversely, a drop of the water table may cause folia to form on a previously
submerged mammillary calcite-coated wall segment. Dating the uppermost mammillary calcite

underneath such folia constrains the age of this water table decline at the respective elevation
(Wendt et al., 2018).

*Figure 1. A: Overview map of North America B: Shaded relief map of the study area showing the Ash Meadows Groundwater*
*Basin, Devils Hole and the two main regions of groundwater recharge (Spring Mountains and Sheep Range). Blue arrows indicate*
*the main direction of groundwater flow. C: Simplified transect of Devils Hole #2 with the position of horizontally drilled cores in*
*the hanging wall of this fracture, providing a long-term record of calcite deposition.*

METHODS

Eleven cores of 2.5 cm diameter and 0.1-1.3 m length were drilled horizontally into the calcite
deposits of the hanging wall of Devils Hole #2 at elevations between +3.2 m and +19.5 m
relative to the modern water table. The cores were cut in half and polished. Calcite that appears
macroscopically white was associated with folia (see Supplementary material for additional
information on folia and its subtype referred to as proto-folia). In each core, samples of
mammillary calcite adjacent to folia were sampled for U-series dating. Mammillary calcite
samples taken immediately (<5 mm) underneath a folia layer are categorized to mark the
decrease of the water table. Conversely, mammillary calcite samples just above (<5 mm) a folia
layer are categorized to record an increase in water table. Samples of thin (<10 mm) mammillary
calcite layers bracketed by folia are attributed to brief rises of the water table. Mammillary
calcite samples more than 5 mm from any folia are categorized to mark submergence. This
approach is similar to the one used by Wendt et al. (2018). Samples for dating (20-50 mg) were
drilled perpendicular to the growth axis (Figures S1 and S2) and two aliquots (0.2-0.3 mg each)
were used for stable isotope analyses.

U-series dating was performed at the University of Minnesota (USA). Samples were digested in
HNO₃ and spiked with a mixed ²³³U-²³⁶U-²²⁹Th spike similar to that described in Edwards et al.
(1987). Spiked samples were co-precipitated with Fe, centrifuged, and loaded into anion

exchange columns following the methods described by Shen et al. (2002, 2012). Separate
uranium and thorium liquid extracts were measured on an inductively coupled plasma mass
spectrometer (Thermo Neptune Plus) via a secondary electron multiplier using a peak-jumping
mode (Shen et al., 2012; Cheng et al., 2013). Ages were calculated using the ^{230}Th and ^{234}U half-
lives of Cheng et al. (2013). Chemical blanks were measured with each set of 12 samples and
were found to be negligible (<50 ag ^{230}Th , <150 ag ^{234}U , <150 fg ^{232}Th , <1 pg ^{238}U). In order to
derive ^{234}U -U ages, $\delta^{234}\text{U}_{\text{initial}}$ was calculated from stable oxygen and carbon isotopes using the
regression equations of Li et al. (2020). Its exponential decay to $\delta^{234}\text{U}_{\text{measured}}$ constrains the age.
All uncertainties are given as 2σ ranges.

Oxygen and carbon isotope analyses were performed at the University of Innsbruck. The sample
powders were analyzed using a semi-automated device (Gasbench II) linked to an isotope ratio
mass spectrometer (ThermoFisher Delta V). Isotope values are reported relative to VPDB. Long-
term precision is better than 0.1‰ for both $\delta^{13}\text{C}$ and $\delta^{18}\text{O}$ (Spötl and Vennemann 2003). The
carbon and oxygen values of the duplicates from each sample replicated typically within $\pm 0.1\%$,
never exceeding 0.2‰, and the mean value was taken.

RESULTS

173 ^{230}Th -U and ^{234}U -U chronologies

For each measured sample there are two possible approaches for calculating an age. ^{230}Th -U ages
(Kaufman and Broecker, 1965) can be used for samples up to 650 ka; for older samples the
uncertainties become exceedingly large (Figure 2). For Devils Hole, the relationship between
$\delta^{234}\text{U}_{\text{initial}}$ and $\delta^{13}\text{C}$ and $\delta^{18}\text{O}$ (Li et al., 2020) was used to also calculate ^{234}U -U ages based only
on the decay of ^{238}U to ^{234}U . In total, 92 samples were processed in the range of 350 to 820 ka
(Supplementary Material). For 59 samples, where both ages could be derived (i.e., they are
within the physical limits of ^{230}Th -U dating, defined here by a ^{230}Th -U age uncertainty < 1
million years), the two datasets agree within uncertainty (Figure 2A).

Samples deposited around 450 ka display a similar uncertainty for both age calculation methods
(Figure 2B). For the purpose of reconstructing the water table record, all samples with a ^{230}Th -U
age > 450 ka are represented by their ^{234}U -U age instead.

Five out of the 92 samples (samples N-754, N-749, J-813, F-573 and G-426) were found to be
significantly out of micro-stratigraphic order and were not used (see Supplementary Materials for
additional information).

The quoted age uncertainties incorporate the statistical uncertainty of ion counting and a
systematic uncertainty. For the calculation of ^{230}Th -U ages, the statistical uncertainty is apparent
in the uncertainty of the ‘uncorrected Th-age’ and a systematic uncertainty contribution from
detrital ^{230}Th is mostly negligible, owing to the age of the samples and the low concentrations of
192 ^{232}Th , which serves as a proxy for detrital ^{230}Th (measured atomic ratio $^{230}\text{Th}/^{232}\text{Th} > 9.91 \cdot 10^{-4}$).
For the calculation of ^{234}U -U ages, the statistical uncertainty is sourced in the $\delta^{234}\text{U}_{\text{measured}}$ and
is small compared to the systematic uncertainty from assessing the $\delta^{234}\text{U}_{\text{initial}}$. The 2σ uncertainty
of $\delta^{234}\text{U}_{\text{initial}}$ in Devils Hole calcite was calculated by Li et al. (2020) to be 60.5‰ for all samples.
For 78 samples (85%) ^{230}Th -U and ^{234}U -U ages agree within their 2σ uncertainties and only four

Commented [A2]: Suggest a Xref to supp material and just state there were 5 outliers.

samples (4%) have a difference of more than 1.5x their combined uncertainty. Overall the results
of both methods show a good agreement.

Construction of water-table changes

The oldest mammillary calcite deposition is recorded in four cores between +4.6 m and +8.3 m
elevation and dates to 820 ka (mean of samples Ibis825, N771, N775, J992 and G867). Apart
from this 820 ka layer, the next youngest mammillary calcite formed around 750 ka and is
present at all sampled elevations above 4 m. The water table decline after the brief rise at 746±16
205 ka ago at (+19.5 m) marks the start of the continuous record of the water-table changes (Figure
3).

Water-table oscillations prior to 350 ka reached up to 15 m in amplitude and their rate was
comparable to the rate of water-table drops during Terminations in the last 350 ka (Szabo et al.,
1994; Wendt et al., 2018). Furthermore, our record reveals smaller amplitude oscillations in
between which show periods of roughly 20 kyr. Mammillary calcite deposition at the highest
elevations (>15 m) only occurred prior to 600 ka, while mammillary calcite layers in cores
around 8-10 m elevation are thicker in the older part and become thinner in more recent glacial
periods.

Long-term water-table decline component

The youngest ages of mammillary calcite obtained from each of the five highest cores are
become older the higher as the elevation of the core increases: at +19.5 m the last deposition
dates to 719±14 ka, at +18.7 m to 660±15 ka, at +15.8 m to 643±15 ka, at +13.3 m to 567±14 ka.
The core at +9.9 m contains a last thick (>1cm) mammillary calcite layer that formed at 438±14
219 ka. Four out of these five mammillary calcite layers (all but the one at +15.8 m) are overgrown
by folia or proto-folia indicating a water-table decline immediately after their deposition. Con-
sidering an age of 719 ka at +19.5 m and 438 ka at +9.9 m suggests a mean rate of water-table
decline of 3.4 m per 100 kyr on which shorter (<100 kyr) water-table variations are
superimposed. The end of calcite deposition at elevations of +18.7 m, +15.8 m and +13.3 m is
roughly consistent with a long-term linear component of a declining water table over this time
period (Figure 3). Such a long-term trend is not apparent after 438±14 ka (Wendt et al., 2018).

Lower limit of water-table changes

The identification of long (>100 kyr) periods of continuous growth of mammillary
calcite provides a lower limit of water-table lowstands. Although the absence of folia may be
considered an indicator of continuous submergence, folia may not necessarily form when the
water table drops or rises quickly. Another indication of continuous growth (i.e. the absence of
significant hiatuses) is a constant mean growth rate of 0.9±0.3 mm/kyr found in a previous
study of mammillary calcite from an elevation of +1.8 m (Li et al., 2020).

The core at +5.6 m is interpreted to have been continuously submerged from 682±16 ka where
we have a marker for a rising water table in the core above until 555±13 (see Supplementary
Material Figure S2 and S4). This proto-folia layer was dated in the core above (+6.8m) and
demonstrates submergence after 682±16 ka. The average growth rate of 0.8 mm/kyr as
calculated agrees with the previously published rate of 0.9±0.3 mm/kyr (Li et al., 2020).

Another period of continuous growth at +5.6 m elevation lasted from 532±14 ka (J784) to
418±11 ka (J664). Its average growth rate of 1.1 mm/ka_{yr} also agrees with the published growth
rate (Li et al., 2020). Both growth layers are, within their uncertainties, consistent with sampled
deposits below this elevation (i.e. there was no folia deposition below +5.6 m during these two
intervals).

*Figure 2. A: Comparison of ^{230}Th -U and ^{234}U -U age dates for the 59 samples from Devils Hole where both ages were obtained.*
*For the reconstruction of the water-table record (Figure 3), the ^{234}U age was used for samples with a ^{230}Th -U age older than 450*
*ka (vertical line). B: Comparison of uncertainties of 53 ^{230}Th -U and 86 ^{234}U -U ages. ^{230}Th -U ages older than 450 ka (vertical line)*
*were not used for the water-table reconstruction (Figure 3) and were substituted by their ^{234}U -U age.*

*Figure 3. Elevation of the paleo-water table in Devils Hole relative to today's water table position (0 m), based on this study,*
*Wendt et al. (2018) and Szabo et al. (1994). The black dashed line is the long-term component of decline. Our interpretation of*
*water-table changes across the last 750 ka (based on Szabo et al., 1994, and Wendt et al., 2018) and this study is shown by the*
*blue shading. Dense blue color means "submerged with high certainty", white means "not submerged with high certainty" and*
*intermediate colors are a qualitative measure of the likeliness of submergence. Intermediate blue shading reflects lack of data,*
*dating uncertainty and the fact that due to the slow growth of mammillary calcite, changes on timescales <10 kyr cannot be*
*resolved. No interpretation of the water table is provided for the oldest part of the record prior to 750 ka due to the scarcity of the*
*data.*

DISCUSSION

Limitations

Since the boundary between folia and mammillary calcite cannot be directly dated, ages for
declining or increasing water table, as derived from mammillary calcite a few mm distant from
the respective boundaries, are systematically too old or too young by up to 5 kyr respectively. In
this study, no attempt was made to extrapolate the age to the boundary between mammillary
calcite and folia by measuring multiple ages (cf. Wendt et al., 2018) because the uncertainties of
these older parts of the deposit precluded such an approach. Sampling-derived offsets, however,
are negligible compared to other uncertainties and do not affect the interpretation.

Given the dating uncertainty it is not possible to resolve changes that occur on timescales smaller
than 10 kyr. The blue shading in Figure 3 is our interpretation of changes on timescales >10
275 kyr. Some of the cores show many oscillations between thin layers of mammillary calcite and
276 folia, suggesting higher-frequency water-table fluctuations than shown in Figure 3. Such high-
277 frequency fluctuations are especially prevalent during the 30 to 40 kyr-long highstands (+18.7
278 m and +19.5 m) at 660 ka and 740 ka.

Water-table elevation changes in the Ash Meadows aquifer upstream of Devils Hole

Wendt et al. (2020) found that $\delta^{234}\text{U}_{\text{initial}}$ variations captured by the calcite reflect water-table
changes in the aquifer upstream of Devils Hole. $\delta^{234}\text{U}_{\text{initial}}$ is expected to increase simultaneously
with the rise in water table following long and deep lowstands as a result of the submergence of
rock (and associated water-rock interactions) previously located in the unsaturated zone (Wendt
et al., 2020). Wendt et al. (2020) discussed different sources of $\delta^{234}\text{U}_{\text{initial}}$ variations in Devils
Hole and found that processes other than the discussed effect of water table elevation to be
negligible over the past 350 kyr. It is assumed here that this conclusion can be extended back to
750 ka. Therefore, a water table rise after a major lowstand in Devils Hole that was not
accompanied by a rise in $\delta^{234}\text{U}_{\text{initial}}$ indicates a local origin of the water table change that did not
affect the whole aquifer and was thus not linked to climate change.

In the time period between 820 ka and 350 ka only one significant period of low $\delta^{234}\text{U}_{\text{initial}}$ dated
at around 410 ka was followed by an increase in $\delta^{234}\text{U}_{\text{initial}}$ (Figure 4). Similar rises in $\delta^{234}\text{U}_{\text{initial}}$,

[revised manuscript text omitted]

438 ± 14 ka (cf. also Wendt et al., 2018). The rate of the long-term decline is smaller than ~~that~~
suggested by Winograd and Szabo (1988), because about half of the 26 m decline after 750 ka
was caused by climate-induced changes. The model of a slowed decline (Winograd and Szabo,
1988) is consistent with this study and Wendt et al. (2018).

Impact of Terminations on the Devils Hole water table

Nine glacial-interglacial cycles with corresponding Terminations occurred during the last 820
~~kyra~~ (Railsback et al., 2015). The global impact of Terminations is apparent in sea-level rises
(Spratt and Lisiecki, 2016), shifts to higher atmospheric CO₂ levels (Bereiter et al., 2015) and
intervals of weak Asian monsoon activity (Cheng et al., 2016). Wendt et al. (2018) dated the
youngest four Terminations at Devils Hole, and our study provides data back to Termination IX.

The Devils Hole data document a strong decline in water table between 438 ± 14 ka and 403 ± 6 ka
from more than +9.9 m to less than +3.2 m, associated with Termination V, as well as a
minimum in $\delta^{234}\text{U}_{\text{initial}}$ as expected from a major groundwater lowstand (Wendt et al., 2019). This

timing is consistent with ~~a~~ major rises in atmospheric CO₂ (Bereiter et al., 2015) and ~~a major rise~~
~~in~~ global sea level (Spratt and Lisiecki, 2016) (Figure 5A+B). The Asian monsoon was weak
between 430.5±1.5 ka and 426±2 ka, which agrees with the early phase of water-table decline at
Devils Hole (Figure 5D).

The water-table lowstand around 550 ka (± 15 ka) predates Termination VI compared to other
climate archives (Bereiter et al., 2015; Cheng et al., 2016; Spratt and Lisiecki, 2016; Railsback et
al., 2015) and was attributed above to local (tectonic) processes that were not linked to changes
in recharge. The corresponding weak Asian monsoon interval ~~was lasted~~ about 4.5 ~~kyra long~~ and
centered at 532.3±3.5 ka (Cheng et al., 2016), which significantly predates the Devils Hole water
table decline at 515±10 ka (mean of samples H374 and H380) recorded only in the core at +9.9
405 m. Considering that the water-table decline associated with Terminations usually extends over
406 20-30 ka (Wendt et al., 2018), the decline at 515±10 ka may reflect the end of what would have
407 been a longer decline period without the 550 ka non-climatic event. Continuous calcite
deposition throughout this time at +5.6 m elevation indicates an interglacial lowstand above this
elevation, higher than for any younger Termination. The magnitude of climatic change
associated with this Termination can thus be considered as rather low. Similarly, a low
magnitude of change at Termination VI is also observed in sea level (Spratt and Lisiecki, 2016;
Figure 5B), atmospheric CO₂ (Bereiter et al., 2015; Figure 5A) and ~~the~~ Asian monsoon records
(Cheng et al., 2016; Figure 5D).

Termination VII is reflected in a water-table decline between 660±15 ka and 627±15 ka from
more than 18.7 m to just below 6.8 m. Although small compared to their uncertainty, $\delta^{234}\text{U}_{\text{initial}}$
values also show a minimum at this time (Figure 4), adding further evidence that this lowstand
affected the whole aquifer. The timing is shifted towards older ages compared to the global sea
level but still within uncertainty and is in good agreement with the Antarctic CO₂ record (Figure
5). The weak Asian monsoon interval ended at 627±6 ka (Cheng et al., 2016) which coincided
also with the end of water-table decline in Devils Hole. Another weak Asian monsoon interval
around 585 ka was associated with Termination VIIa. Our record shows a corresponding
decrease at 573±14 ka from above 13.3 m to below 6.8 m at 584±14 ka.

Termination VIII is reflected by a decline after brief rises at 19.5 m at 719±14 ka and at 18.7 m
at 724±15 ka and a lowstand at 15.8 m after 720±14 ka. All other cores show a 3 mm-thin layer
of proto-folia (Supplementary Material Figure S4), which was dated to between 722±16 ka and
690±16 ka at 6.8 m. Hence the water-table decline may have had an amplitude of more than 10
427 m, but did not last long enough below 15.8 m to form a thicker folia or proto-folia layer. $\delta^{234}\text{U}$ -
428 initial values show a minimum comparable to their uncertainty (Figure 4) indicating a lowstand,
although with low certainty.

Termination IX falls into the gap of the earliest deposition at 820 ka and the onset of the
continuous water-table record around 750 ka. The $\delta^{234}\text{U}_{\text{initial}}$ values also cannot resolve an
increasing trend after this Termination which would correspond to a lowstand of high magnitude
and long duration due to the high uncertainties.

Although uncertainties of 60.5‰ for $\delta^{234}\text{U}_{\text{initial}}$ are about one third of the total observed
variability (between 1650-1850 ‰) and considering their indirect measurement via stable
oxygen and carbon isotopes, it is worth noting that Terminations before the Mid-Brunhes event
(i.e. Termination VI and older) were not accompanied by a shift in $\delta^{234}\text{U}_{\text{initial}}$ over 200‰, as seen
for Terminations II, IV (Wendt et al., 2020) and V. A smaller shift in $\delta^{234}\text{U}_{\text{initial}}$ could imply that

the interglacials following these Terminations were not arid or were rather short, or a
combination of both.

Insolation-induced ice-volume changes in North America as driver of regional moisture
availability during moderately warm climate states

Between 630 ka and 450 ka, the global climate was moderately warm, as seen by with sea-level
variations between -10 m and -70 m relative to today (Spratt and Lisiecki, 2016) and a CO₂ level
below 260 ppm (Bereiter, 2015). During this time, the Devils Hole water table exhibited multiple
oscillations with a cyclicity of about 20-30 kyra and amplitudes smaller than during most
Terminations. Another oscillation occurred around 700 ka following Termination VIII. These
oscillations align well with the 65°N mid-July insolation (Figure 5C) that has a cyclicity mode of
about 22 kayr (Berger, 1999).

The connection between water table and insolation follows the rationale that high water tables
are a proxy for increased regional P-ET (precipitation minus evapotranspiration). P-ET increases
due to cooler temperatures (Santi et al., 2020) and, most importantly, due to southerly shifted
storm tracks that deliver additional moisture. Paleoclimate simulations (Oster et al., 2015) show
that extensive northwestern American ice sheets shift winter storm tracks further south and into
the Great Basin, due to the development of stable high-pressure systems over these ice sheets.
The growth and decay of the Cordilleran ice sheet in northwestern North America has shown to
have been exceptionally sensitive to these insolation changes during phases of moderate climate,
as seen during Marine Isotope stage 13 (Niu et al., 2021), which lasted from 533 ka to 478 ka
(Lisiecki and Raymo, 2005). The water-table oscillations identified in this study underscore that
the mechanism responsible for wetting the Great Basin does not only operate between glacial and
interglacial climates but is also sensitive to ~~also~~ smaller changes on suborbital timescales. The
extent of the Laurentide ice sheet during MIS 13 is not well known but it was likely small or
even non-existent (Bailey et al., 2010; Niu et al., 2021). This would suggest that, at least during
climates like during MIS 13, high-pressure systems over the Cordilleran ice sheet played a key
role in steering winter moisture into the Great Basin. It is acknowledged that temperature
changes following the insolation changes influence P-ET and hence also affect recharge
dynamics and hence the water-table elevations. It remains to be quantified if this effect alone
may have driven water-table changes during MIS 13 and similar climate states.

*Figure 5. Comparison between the Devils Hole water-table evolution and (A) atmospheric CO₂ from Antarctic ice cores (Bereiter,*
*2015), (B) global sea level (Spratt and Lisiecki, 2016; note inverted Y axis), (C) mid-July 65°N insolation (Berger, 1999), and (D)*
*the Asian monsoon (Cheng et al., 2016). Roman numbers refer to Terminations (Railsback et al., 2015) as seen in major shifts*
*towards higher CO₂ values or higher sea levels. Blue shading in the background is the interpreted qualitative probability of water*
*table height between 750 ka and 0 ka based on the data shown in Figure 3.*

ACKNOWLEDGMENTS

This research was funded by grant P327510 of the Austrian Science Fund. Fieldwork was
conducted under the scientific research and collecting permits of the United States National Park
Services DEVA-2022-SCI-0014, DEVA-2017-SCI-0002 and DEVA-2015-SCI-0006.

AUTHOR CONTRIBUTIONS

SDS contributed with design of the study, investigations, formal analysis and writing the original
draft. KW contributed with conceptualization, investigations and reviewing the draft. YD
contributed with investigations, supervision and reviewing the draft. RLE contributed with
infrastructure and reviewing the draft. XL contributed with formal analysis and reviewing the draft.
GMC contributed with formal analysis. GEM contributed with conceptualization and reviewing
the draft. CS contributed with conceptualization, investigations, funding acquisition, supervision,
infrastructure and reviewing the draft. All authors contributed to the final version of the
manuscript.

DECLARATION OF INTERESTS

The authors declare no competing interests.

DATA AVAILABILITY

All data are released with this study as supplemental material. Additionally they are in the
process of being released on PANGAEA.

SUPPLEMENTAL INFORMATION

The supplemental information includes one text file with figures and tables as well as two data
files.

REFERENCES

- Anderson, R.E., Crone, A.J., Machette, M.N., Bradley, L.A., & Diehl, S.F. Characterization of
Quaternary and suspected Quaternary faults, Amargosa area, Nevada and California
USGS-OFR-95-613 (1995)
- Audra et al., 2009: Audra Ph., Mocochain L., Bigot J.-Y. and Nobécourt J.-C. 2009. The
association between bubble trails and folia: a morphological and sedimentary indicator of
hypogenic speleogenesis by degassing, example from Adaouste Cave (Provence, France).
*International Journal of Speleology*, 38 (2), 93-102. Bologna (Italy). ISSN 0392-6672.

Bailey et al., 2010 A low threshold for North Atlantic ice rafting from “low-slung slippery” late
Pliocene ice sheets <https://doi.org/10.1029/2009PA001736>
Bajnai et al., 2021 Devils Hole Calcite Was Precipitated at $\pm 1^{\circ}\text{C}$ Stable Aquifer Temperatures
During the Last Half Million Years <https://doi.org/10.1029/2021GL093257>
Berger, A; Loutre, M-F (1999): Parameters of the Earths orbit for the last 5 Million years in 1 kyr
resolution. PANGAEA, <https://doi.org/10.1594/PANGAEA.56040>
Bhattacharya et al (2017) Glacial reduction of the North American Monsoon via surface cooling
and atmospheric ventilation <https://doi.org/10.1002/2017GL073632>
Bereiter, B et al (2015) Revision of the EPICA Dome C CO₂ record from 800 to 600 kyr before
present <https://doi.org/10.1002/2014GL061957>
Cheng, H. *et al.* Improvements in ²³⁰Th dating, ²³⁰Th and ²³⁴U half-life values, and U–Th
isotopic measurements by multi-collector inductively coupled plasma mass spectrometry.
Earth and Planetary Science Letters. **371-372**, 82–91
<https://doi.org/10.1016/j.epsl.2013.04.006> (2013).
Cheng, H. et al., “The Asian monsoon over the past 640,000 years and ice age terminations,”
Nature, vol. 534, no. 7609, pp. 640–646, Jun. 2016, <https://doi.org/10.1038/nature18591>
Conroy et al., 2019 Surface winds across eastern and midcontinental North America during the
Last Glacial Maximum: A new data-model assessment
<https://doi.org/10.1016/j.quascirev.2019.07.003>
Cuttillo and Ge, 2006 Analysis of strain-induced ground-water fluctuations at Devils Hole, Nevada
<https://doi.org/10.1111/j.1468-8123.2006.00150.x>
D’Angeli et al., 2015: Genesis of folia in a non-thermal epigenic cave (Matanzas, Cuba)
<http://dx.doi.org/10.1016/j.geomorph.2014.09.006>
Davis 2012: In defense of a fluctuating-interface, particle-accretion origin of folia
<http://dx.doi.org/10.5038/1827-806X.41.2.6>
Deacon, James E., and C. Deacon Williams. "Ash Meadows and the legacy of the Devils Hole
pupfish." Battle against extinction: native fish management in the American West (1991):
69-87.
Deacon et al (2007) Fueling Population Growth in Las Vegas: How Large-scale Groundwater
Withdrawal Could Burn Regional Biodiversity <https://doi.org/10.1641/B570809>
Dublyansky and Spötl 2015 Condensation-corrosion speleogenesis above a carbonate-saturated
aquifer: Devils Hole Ridge, Nevada <http://dx.doi.org/10.1016/j.geomorph.2014.03.019>
Edwards, R. L., Chen, J. H. and Wasserburg, G. J. ²³⁸U-²³⁴U-²³⁰Th-²³²Th systematics and the
precise measurement of time over the past 500,000 years. Earth and Planetary Science
Letters. **81**, 175–192 [https://doi.org/10.1016/0012-821X\(87\)90154-3](https://doi.org/10.1016/0012-821X(87)90154-3) (1987).
EPICA members (2004) Eight glacial cycles from an Antarctic ice core
<https://doi.org/10.1038/nature02599>
Halford, K. J. & Jackson, T. R. Groundwater characterization and effects of pumping in the Death
Valley regional groundwater flow system, Nevada and California, with special reference
to Devils Hole. U.S. Geological Survey Professional Paper 1863.
<https://doi.org/10.3133/pp1863> (2020).
Jackson and Steidle et al., 2023 A 350,000-year history of groundwater recharge in the southern
Great Basin, USA <https://doi.org/10.1038/s43247-023-00762-0>
Kaufman, A. and Broecker, W. August 1965. Comparison of Th²³⁰ and C¹⁴ ages for carbonate
materials from lakes Lahontan and Bonneville. Journal of Geophysical Research,
70(16):40394054. ISSN 0148-0227. doi:10.1029/jz070i016p04039.

Kluge et al., 2014 Devils Hole paleotemperatures and implications for oxygen isotope equilibrium
fractionation <http://dx.doi.org/10.1016/j.epsl.2014.05.047>
Kolesar, P. T. and A. C. Riggs, “Influence of depositional environment on Devils Hole calcite
morphology and petrology,” in Studies of Cave Sediments, Ed.I. D. Sasowsky and J.
Mylroie Springer US, 2004, pp. 227–241.
Li, X., Wendt, K. A., Dublyansky, Y. *et al.* Novel method for determining 234U-238U ages of
Devils Hole 2 cave calcite. *Geochronology* <https://doi.org/10.5194/gchron-2020-26>
(2020).
Lisiecki, L. E. and M. E. Raymo, “A Pliocene-Pleistocene stack of 57 globally distributed benthic
18O records,” *Paleoceanography*, vol. 20, no. 1, p. n, Jan. 2005, doi:
10.1029/2004pa001071.
Lora et al., 2017 North Pacific atmospheric rivers and their influence on western North America
at the Last Glacial Maximum <https://doi.org/10.1002/2016GL071541>
Lopez Martinez et al, 2020: Bubble trail and folia in cenote Zapote, Mexico: petrographic evidence
for abiotic precipitation driven by CO2 degassing below the water table
<https://doi.org/10.5038/1827-806X.49.3.2344>
Lyle, M. *et al.* Out of the Tropics: The Pacific, Great Basin Lakes, and Late Pleistocene Water
Cycle in the Western United States. *Science* **337** 1629–1633
<https://doi.org/10.1126/science.1218390> (2012).
MacDonald, G. M. Water, climate change, and sustainability in the southwest. *Proc. Natl. Acad.*
*Sci.* **107**, 50, 21256-21262; 10.1073/pnas.0909651107 (2010).
Moseley G. E. *et al.* Reconciliation of the Devils Hole climate record with orbital forcing. *Science*
**351** 165–168 <https://doi.org/10.1126/science.aad4132> (2016).
Niu 2021; Coupled climate-ice sheet modelling of MIS-13 reveals a sensitive Cordilleran Ice Sheet
<https://doi.org/10.1016/j.gloplacha.2021.103474>
Oster, J. L., Ibarra, D. E., Winnick, M. J. & Maher, K. Steering of westerly storms over western
North America at the Last Glacial Maximum. *Nat. Geosci.* **8**, 201-205 (2015).
Pérouse, E. and Wernicke, B. P. Spatiotemporal evolution of fault slip rates in deforming
continents: The case of the Great Basin region, northern Basin and Range province
*Geosphere*, vol. 13, no. 1, pp. 112–135, Nov. 2016, doi: 10.1130/ges01295.1.
Plummer, L.N., E. Busenberg, and A. C. Riggs, “In-situ Growth of Calcite at Devils Hole, Nevada:
Comparison of Field and Laboratory Rates to a 500,000 Year Record of Near-Equilibrium
Calcite Growth,” *Aquatic Geochemistry*, vol. 6, no. 2, pp. 257–274, 2000, doi:
10.1023/a:1009627710476.
591 L. B. Railsback, P. L. Gibbard, M. J. Head, N. R. G. Voarintsoa, and S. Toucanne, “An optimized
scheme of lettered marine isotope substages for the last 1.0 million years, and the
climatostratigraphic nature of isotope stages and substages,” *Quaternary Science Reviews*,
594 vol. 111, pp. 94–106, Mar. 2015, doi: 10.1016/j.quascirev.2015.01.012.
Reheis, M. C.; Caskey, J.; Bright, J.; Paces, J. B.; Mahan, S. & Wan, E. Pleistocene lakes and
paleohydrologic environments of the Tecopa basin, California: Constraints on the drainage
integration of the Amargosa River *GSA Bulletin*, Geological Society of America, 2019
Santi et al., 2020 Clumped isotope constraints on changes in latest Pleistocene hydroclimate in the
northwestern Great Basin: Lake Surprise, California <https://doi.org/10.1130/B35484.1>
Seltzer et al., 2019 Deglacial water-table decline in Southern California recorded by noble gas
isotopes <https://doi.org/10.1038/s41467-019-13693-2>

Shen, C.-C. *et al.* Uranium and thorium isotopic and concentration measurements by magnetic
sector inductively coupled plasma mass spectrometry. *Chemical Geology*, **185**, 165–178
[https://doi.org/10.1016/S0009-2541\(01\)00404-1](https://doi.org/10.1016/S0009-2541(01)00404-1) (2002).

Shen, C.-C. *et al.* High-precision and high-resolution carbonate ²³⁰Th dating by MC-ICP-MS
with SEM protocols. *Geochimica et Cosmochimica Acta*. **99**, 71–86
<https://doi.org/10.1016/j.gca.2012.09.018> (2012).

Spötl, C. and Vennemann, T. W. Continuous-flow isotope ratio mass spectrometric analysis of
carbonate minerals. *Rapid Communications in Mass Spectrometry*. **17**, 1004–1006
<https://doi.org/10.1002/rcm.1010> (2003).

Spratt, Rachel M. and Lorraine E. Lisiecki (2016) A Late Pleistocene sea level stack
<https://cp.copernicus.org/articles/12/1079/2016/>

Stein et al (2009) Variability of surface water characteristics and Heinrich-like events in the
Pleistocene midlatitude North Atlantic Ocean: Biomarker and XRD records from IODP
Site U1313 (MIS 16–9) <https://doi.org/10.1029/2008PA001639>

Szabo et al (1994) Paleoclimatic Inferences from a 120,000-Yr Calcite Record of Water-Table
Fluctuation in Browns Room of Devils Hole, Nevada
<https://doi.org/10.1006/qres.1994.1007>

Wendt, K. A. *et al.* Moisture availability in the southwest United States over the last three glacial-
interglacial cycles. *Science Advances*. AAAS. **4** <https://doi.org/10.1126/sciadv.aau1375>
(2018).

Wendt et al. Paleohydrology of southwest Nevada (USA) based on groundwater ²³⁴U/²³⁸U over
the past 475 k.y. *GSA Bulletin* 132 (3-4): 793–802. (2020)
<https://doi.org/10.1130/B35168.1>

Winograd, I. J., Riggs, A. C. & Coplen, T. B. The relative contributions of summer and cool-
season precipitation to groundwater recharge, Spring Mountains, Nevada, USA. *Hydrogeol*
*J.* Springer Science and Business Media LLC, **6**, 77-93 (1998).

Winograd, I. J. and Szabo, B. J. (1988) Water-table decline in the south-central Great Basin during
the Quaternary: Implications for toxic waste disposal In: “Geologic and Hydrologic
Investigations of a Potential Nuclear Waste Disposal Site at Yucca Mountain, Southern
Nevada” (M.D. Carr and J. C. Younts, Eds.) U.S. Geological Survey Bulletin 1790, pp.
147-152 doi: 10.2172/60629

In-Depth Review of Steidle et al., 2023 entitled 'Moisture availability and groundwater recharge paced by orbital forcing over the past 750,000 years in the southwestern USA'

The following review is written in chronological order with the aim of strengthening the arguments and addressing uncertainties or lack thereof within the manuscript.

The Abstract is clear and concise and communicates the main results and conclusions. I would recommend that in L28, the authors state by providing a numerical number or range for the 'older half of the record' where the long-term decline is posited.

L44-45, it is recommended that authors use updated and more recent references that detail extensive work being done in southwestern US with respect to groundwater variability:

Miller, O.L., Putman, A.L., Alder, J., Miller, M., Jones, D.K. and Wise, D.R., 2021. Changing climate drives future streamflow declines and challenges in meeting water demand across the southwestern United States. *Journal of Hydrology X*, 11, p.100074.

Masbruch, M.D., Rumsey, C.A., Gangopadhyay, S., Susong, D.D. and Pruitt, T., 2016. Analyses of infrequent (quasi-decadal) large groundwater recharge events in the northern Great Basin: Their importance for groundwater availability, use, and management. *Water Resources Research*, 52(10), pp.7819-7836.

MacDonald, G.M., 2010. Water, climate change, and sustainability in the southwest. *Proceedings of the National Academy of Sciences*, 107(50), pp.21256-21262.

Meixner, T., Manning, A.H., Stonestrom, D.A., Allen, D.M., Ajami, H., Blasch, K.W., Brookfield, A.E., Castro, C.L., Clark, J.F., Gochis, D.J. and Flint, A.L., 2016. Implications of projected climate change for groundwater recharge in the western United States. *Journal of Hydrology*, 534, pp.124-138.

Siirila-Woodburn, E.R., Rhoades, A.M., Hatchett, B.J., Huning, L.S., Szinai, J., Tague, C., Nico, P.S., Feldman, D.R., Jones, A.D., Collins, W.D. and Kaatz, L., 2021. A low-to-no snow future and its impacts on water resources in the western United States. *Nature Reviews Earth & Environment*, 2(11), pp.800-819.

Figure 1A: It might be helpful to conceptually draw the major drivers of rainfall over Devils Hole in the modern or during periods of large Ice Sheets such as the Laurentide/Cordilleran over the US. Authors cite papers from Bhattacharya and Oster, which provide conceptual diagrams, during their discussion in the southernly shift of the moisture trajectories.

L108-110; stating that 'the groundwater table shows the same elevation in both caves, suggesting that they are hydraulically connected' warrants a citation/ presentation of the data. I might have missed that if it is presented in the manuscript.

L141-149: It would be helpful to the reader to reference Table S1 here. It is not clear to me why, mammillary calcite <10mm is considered thin and associated with a brief increase in water table? 10mm is thicker than 5mm, which is being associated here as a decrease in water level. Wendt et al., 2018 in their '*Interpretation of thin mammillary calcite layers*' section discuss '*Mammillary calcite (subaqueously deposited) intervals (≤ 0.5 mm in width)*'

Figure 2 and associated text in L172-196, might help to underline the difference between the 53 ^{230}Th -U and 86 ^{234}U -U. IF there are a total of 92 samples that were analyzed and 5 were out of stratigraphic order, which gives 87 samples, what is the sampling distribution of the 86 ages.

L200 authors use IDs of the samples but in other places, they interchange with the depths associated with the cores, for example, L214. It might be helpful to be consistent. As the focus of the manuscript is associated with water level depths, I would suggest that the authors use numerical values and add in parentheses the core IDs.

L200, it is unclear if the oldest mamillary calcite date of 820 ka is the average taken from the 5 cores, Ibis825, N771, N775, J992, and G867. If it is an average of the cores, it would be helpful to mention that and assign an uncertainty. Similarly, it is stated in L201, '750 ka is present at all sampled elevations above 4m.' It is unclear whether this is an average or a single date from a single core.

L212 missing word after older 'than'

The following line states the cores but, in this case, only mentions the heights, which is more important, I agree, but to be consistent with L200, it would help to add core IDs/

L219: I might have missed why <100 ka is considered shorter? Similarly, maybe provide a line of justification on using >100 ka as 'long'

L232: Should Figure S2 instead be S3 as this is the first petrographic image depiction the transition mentioned? Further, it might be helpful to include the petrographic figures more robustly in the main text. Next, Fluid inclusions shown in Figure S3 and S6 are not really discussed in the manuscript and hence I suggest either include them in the discussion or remove them.

L234: When calculating average growth rates and then comparing them with Li et al., 2020, I would recommend adding error bars. Similar with the growth rate calculated in L237.

Figure 3: I appreciate the effort in constructing a hypothesized visual representation of the water table. I would recommend that the authors provide an explanation on what "submerged with high certainty" L254 means. Is that at the 95% CL? How was that calculated? Similarly, provide an explanation for "not submerged with high certainty." Lastly, how was the qualitative measurement done? Was it an interpolation? Adding this to Figure S7 might help in understanding how the shading was developed.

Further, it is interesting to note that the dashed black line highlighting the decline and discussed is evident by only periods of brief (depicted by the diamonds) rise of the water table around 650 and 750 kyrs. What is temporal range constituting this 'brief' period? Is it less than 10 kyrs? But then, authors state in L268-269 that timescales smaller than 10 ka are not resolvable due to dating uncertainty. So I am a little confused here...

Further, the water table with a large number of data points is around 10 m covering 400-750 kyrs. So, I was curious if the authors have thought whether a sampling bias produces the suggested decline?

L28-281: It might be helpful to briefly state what the 'other processes' are and how they are indicative of being negligible to water table elevation.

L286: How was the significance identified in the following, 'only one significant period'? It would also help to identify in Figure 4 by a vertical dashed line when these changes occur.

L288: Could you add a numerical value when 'lower amplitude' is mentioned? Similarly, what is meant by uncertainties larger than the amplitude? Lastly, indicate again Figure 4 when these changes occur.

L293: 'Major changes to the aquifer,' such as...

L309: It is interesting to notice d18O reflected close to glacial values. What are the possible reasons for d13C behaving so differently? A line or two of discussion might be helpful.

L355: 'further past' reads incorrectly.

L388: from should be for

L389: Wend et al., 2019 is not cited.

L409: from should be 'for'

L409 and L417: The authors should address the result that terminations are within error of each other; 660 ± 15 and 627 ± 15 kya. Separately, 573 ± 14 and 584 ± 14 kya.

Further, Termination VIII brief rises and the lowstand is within the <10 ka age uncertainty. Therefore, I am not sure how much weight can be applied to these interpretations.

L442: If authors state 'amplitudes smaller than during most Terminations' it would be helpful to provide approximate numerical values. Smaller compared to what?

L449: A new paleoclimate simulation might be another good citation to add:

Oster, Jessica L., et al. "North Atlantic meltwater during Heinrich Stadial 1 drives wetter climate with more atmospheric rivers in western North America." *Science Advances* 9.46 (2023): eadj2225.

L452: I am curious whether the growth or decay of the Cordilleran ice sheet or the Laurentide Ice Sheet (L458) are more influential to the moisture balance for Devils Hole?

Answers by the authors in green.

Reviewer #1 (Remarks to the Author):

The Devils Hole record is a foundational record of western US hydroclimate, forming one of the longest continuous records that exists over the last 500 ka in this region. It has formed one of the canonical climate records with which other climate records from this region are compared and contrasted, akin to how the Hulu Cave record is utilized in Asian paleoclimate comparisons. Substantial work within the past decade has focused on bringing coherence to the geochronology at Devils Hole, and new and extended age models have been utilized to look in further detail at proxy records of rainfall, local water-table changes, and tectonics among other things. The current study employs predominantly ^{234}U -U dating using $\delta^{234}\text{U}$ initial estimates based on modeled regressions to $\delta^{18}\text{O}$ and $\delta^{13}\text{C}$ data, to extend the existing Devils Hole water table record from 350 to 750 ka ago. The authors interpret this older part of the water table record to be composed of long-term local tectonic forcing and broader-scale hydroclimate forcing. Due to the wide use of the Devils Hole data and its importance to both climate reconstruction comparisons and as a baseline for climate model efforts, this study extending the water table record back 100s of ka and utilizing a relatively new method of extending U-Th dating beyond traditional timescales (using the ^{234}U -U dating method, via modeled estimates of $\delta^{234}\text{U}$ initial) is of interest to readers of *Communications Earth & Environment*.

I found the paper overall to be well written and clearly structured. The inclusion of detailed supplementary data and figures was very helpful to assess the main arguments of the paper. However, I do think that the manuscript could benefit substantially in a few areas from some more rigorous data analysis, and clearer explanation of decisions used when plotting or talking about subsets of the data, and expansion of reasoning behind some key assumptions made in order to reach the stated conclusions. This is crucial as the water table record, and inferences drawn from it throughout the discussion, are underpinned by the data set itself. I recommend that a substantial revision is warranted to deal with the data and assumptions more robustly before this paper is ready for publication and explain comments in more detail below.

Major comments:

1. Robustness of assumptions on $\delta^{234}\text{U}$ initial regression model

The regression equations of Li et al. (2020) are used to translate $\delta^{18}\text{O}$ and $\delta^{13}\text{C}$ measurements into $\delta^{234}\text{U}$ initial values with which many of the ^{234}U -U ages are calculated. How robust is the assumption that the regression can be extended even further beyond the calibration dataset than the original publication? There is very little written in either Li et al., 2020 or in this manuscript to justify pushing this regression deeper in time. I would like the authors to expand on their reasoning that we can assume the modeled relationship between $\delta^{18}\text{O}$ and $\delta^{13}\text{C}$, and $\delta^{234}\text{U}$ initial, holds this far beyond the calibration dataset. In a similar vein and on a related topic, the manuscript also discusses that the conclusions of Wendt et al. (2020) on other processes that could affect $\delta^{234}\text{U}$ initial besides water table elevation can be extended back to 750 ka, with very little discussion or reasoning to justify this assumption.

This should also be expanded upon. Unless thorough justification can be laid out to the contrary, it would seem that a 60 % uncertainty on the $\delta^{234}\text{U}_{\text{initial}}$ is a MINIMUM expected error as this modeled relationship is pushed further from the calibration dataset. This should be clearly stated as a minimum uncertainty estimate in the manuscript, and plotted as such on the figures.

We added a paragraph to the discussion about this assumption:

The assumption that the driving factor of $\delta^{234}\text{U}_{\text{initial}}$ and its correlation with $\delta^{18}\text{O}$ and $\delta^{13}\text{C}$ (i.e., the findings of Li et al., 2021 and Wendt et al., 2020) were constant since 750 ka is a critical aspect of the age model. Li et al. (2021) provided the $\delta^{234}\text{U}_{\text{initial}}$ regression for the ^{234}U -U age model up to 590 ka. Furthermore, Li et al. (2021) considered three independent groupings of data for different time intervals. There was no significant difference found between these time intervals, providing evidence that there was no gradual change over the last 590 kyr and that the mechanism described in Wendt et al. (2020) therefore was of similar significance before 450 ka. We consider it likely that there was also no gradual change for the 160 kyr before that (i.e., between 750 and 590 ka), as the most likely cause for a significant change in $\delta^{234}\text{U}_{\text{initial}}$ would be an unusually strong tectonic event that caused a massive perturbation of the aquifer (i.e., a major change of the flow path(s) or the recharge area). The data provided by Perouse and Wernicke (2016) do not provide evidence of an outstanding tectonic event in this time range, but rather suggest many small events which were not big enough to significantly alter the uranium concentration integrated over the 80 km long flow path from the source region to Devils Hole. Considering the size of the aquifer and the linkage of $\delta^{234}\text{U}_{\text{initial}}$ to water table elevation it seems unlikely that a shift in $\delta^{234}\text{U}_{\text{initial}}$ without a change in water table height occurred. The presented data show no apparent major sudden change in this time range (750 to 590 ka).

The data presented in this study include ^{230}Th -U ages starting at 784 ± 199 ka (the oldest age with an uncertainty < 100 ka is 686 ± 53 ka), which is close to the start of the water-table record (at 750 ka). As presented in the Results section (and Figure 2), these data support the legitimacy of the methods used to construct ^{234}U -U ages over the last 750 kyr.

Extending the concept of Wendt et al. (2020) further back in time is directly related to extending the ^{234}U -U age model, thus we added these sentences in the second section of the discussion:

“It is assumed here that this conclusion can be extended back to 750 ka. This assumption is directly related to the discussion of extending the ^{234}U -U age model back in time with the same supporting arguments and limitations (see paragraph above).”

2. Comparison of ^{230}Th -U ages with ^{234}U -U ages

a. Line 195: The term “1.5x their combined uncertainty” does not convey meaningful information when it comes to this dataset. Consider more robust description/quantification of closeness of fit (see below point).

We expanded the results section about this aspect (see comment below) and prefer to keep this statement.

b. Line 241: Employ a more robust treatment of the 1:1 plot, e.g., bootstrap the data and calculate a range of R² and p values. This will allow quantification of uncertainties on the R² and p values which can be used to assess the strength of the argument in age comparison. Please state the actual p-value (not just a < number), and state the null hypothesis used in calculating it (i.e., that the ages are not concordant).

Rewrote this part of the main manuscript and added sentences:

For 59 samples where both ages could be derived (i.e., they are within the physical limits of ²³⁰Th-U dating, defined here by a ²³⁰Th-U age uncertainty <1 million years; Figure 2A), a bootstrap analysis (100,000 iterations) of their linear relation was performed. For a linear correlation ($234\text{U-U age} = a + b * 230\text{Th-U age}$) the analysis yielded the following parameters: $a = 50 \pm 101$ ka, $b = 0.90 \pm 0.20$, $R^2 = 0.71 \pm 0.16$ and $p\text{-value} = 5.5\text{E-}10 \pm 1.8.0\text{E-}07$ (uncertainties are 2σ standard deviations). The p-value reflects the likeliness for the null-hypothesis of no concordance. Because ²³⁰Th-U ages close to secular equilibrium scatter towards exceedingly high numbers (an effect not balanced by equal scatter at low values), we repeated the analysis for samples with an uncertainty of <100 ka for their ²³⁰Th-U age (n=52). This analysis yielded the following parameters: $a = -9 \pm 99$ ka, $b = 1.02 \pm 0.20$, $R^2 = 0.76 \pm 0.17$ and $p\text{-value} = 2.9\text{E-}10 \pm 3.9\text{E-}08$. Overall, the results of ²³⁰Th-U and ²³⁴U-U dating show good agreement (Figure 2).

Figure 2 and its caption were adapted accordingly.

3. Inclusion/exclusion of samples from the data set for use

a. Line 194: The samples being discussed here in terms of agreement between ²³⁰Th-U and ²³⁴U-U ages appears to be from the total of 92 samples, however it is mentioned in Line 177 that only 59 of the samples could have ages derived from both dating methods (based on set uncertainty criteria). This is confusing to the reader; please make it clear which data you are using and why.

We added a detailed explanation of all subsets of data in the supplementary material, referred to it at several places in the main manuscript and explain in Figure 2 that some data points are not displayed.

We generally followed the rule to concisely state in the main manuscript the criterion of sample selection (e.g., uncertainty <1000 ka for the ²³⁰Th-U method) and the number of data points resulting from this. We partly see the confusion because two methods were applied to the same samples and in some cases only one of the methods yielded a useful result. We included a detailed description to clarify this. We think that this description is best placed in the supplementary material.

New in the supplement:

“Data treatment, selection of outliers and subsets of data for analysis

In total, 92 samples were processed for this study. Five outliers were removed: four (N-754, N-749, J-813 and F-573) out of the five omitted samples were discarded because they are out of stratigraphic order. They are from thin layers bracketed by petrographic boundaries or rest directly on bedrock. These samples are interpreted as younger deposits whose origin is possibly related to small fractures in the calcite deposit which were later filled by calcite during a

subsequent highstand. Sample G-426 was treated as an outlier and omitted because it is too old with respect to the 2-sigma uncertainty compared to G-425 and G-497 (Figure S2).

In the remaining 87 samples, the measurement of $\delta^{18}\text{O}$ and $\delta^{13}\text{C}$ failed in one case (sample N-544) and could not be repeated with similar accuracy leaving this sample without a ^{234}U -U age. Given its young age of less than the 450 ka threshold, this sample is still useful for the water table chronology but not for the comparison of both dating methods or Figure 2B.

The algorithm to derive ^{230}Th -U ages failed in 18 cases because samples are too close to secular equilibrium (entries with 0 ± 0 ka in the supplementary data table). In nine other cases the numerical result had an age uncertainty >1000 ka. These samples are also regarded as being too close to secular equilibrium. This finally left 60 ^{230}Th -U ages, which were used (not considering the five outliers discussed above).

Given the one sample without available ^{234}U -U age (N-544), there are 59 samples where both dating methods could be successfully applied. For the second bootstrap analysis described in the main text, only samples with uncertainties <100 ka were used (i.e., 52 samples)."

b. Line 241: Why are only 53 of the 59 ^{230}Th -U ages being used and 86 of the ^{234}U -U ages? Why do the ^{230}Th -U ages in Panel B plateau at 650 ka when we clearly see older samples in Panel A; if it is simply because they would plot off the chosen axis scale then this should be clearly stated or better show these with a broken scale bar in order to actually let the reader see all of the available data.

Any exclusions, or decisions made on what data to plot or exclude, needs to be more clearly laid out.

See comment above.

Minor comments:

4. Line 15: The letter "c" is missing from the word "correspondence."

Added

5. Line 41: Add space between number and % sign.

Added

6. Line 42: Used UK spelling of "modelling"; if this is meant to be standardized to American use "modeling."

Changed.

7. Line 129: Panel 1B - the blue arrows are showing up quite blurred and hard to tell which direction they point in; I suggest making these show up more sharply. Panel 1C - the location

lines for Cores K and L have a brown colored strike line through for no apparent reason: either explain the reason in the figure caption or remove the strikes. Panel 1C - for the label for Core “Ibis”, this is written as “I-bis” in the supplemental material: please align the naming between these files.

Made minor changes to Figure 1 based on this comment.

8. Line 96: Add space between number and % sign.

Added

9. Line 102: Add space between number and % sign.

Added

10. Line 131: There is a period missing at the end of “Overview map of North America.”

Added

11. Line 165: The comma is missing after “Vennemann” in the reference.

Added

12. Lines 165-167: Add spaces between numbers and ‰ signs.

Added (2x)

13. Line 178: There is an extra space between the “<” and “1.”

Removed

14. Line 179: The language “agree within uncertainty” seems an overreach here. In examining Panel 2A there are many ages which appear to disagree outside of the plotted uncertainty. Softened language along the lines of “mostly agree within uncertainty” would seem more reasonable.

Added “mostly”

15. Line 182: There is an extra space between the “>” and “450.”

Removed

16. Line 190: There is an extra space between the “>” and “9.91.”

Removed

17. Lines 193-195: Add spaces between numbers and % or ‰ signs.

Added (3x)

18. Line 209: Specify, perhaps in terms of relative %, more precisely what the “thicker” and

“thinner” variations mentioned are, i.e., are mammillary calcite layers 50 % thinner in recent glacial periods?

Providing numbers would require some rough estimates due to the non-binary nature of the calcite deposits and it is not always possible to distinguish between mammillary calcite and folia, especially in the younger part which, however, was not the topic of this study.

We added:

Mammillary calcite deposition at the highest elevations (>15 m) only occurred prior to 600 ka, while mammillary calcite layers in cores at 8-10 m elevation become thinner in more recent glacial periods (cf. longer and more frequent periods of submergence at these elevations in the older part as shown in Figure 3).

19. Line 228: The comma is missing after “i.e.”

Added

20. Line 232: The unit ka is missing after “555±13.”

Added “ka”

21. Line 233: The space is missing between “+6.8” and “m.”

Added

22. Line 239: The comma is missing after “i.e.”

Added

23. Line 241: State whether age uncertainties plotted in Panel A are 1 or 2 sigma on key or in caption, and on the Panel B y-axis.

Added “All age uncertainties are 2σ” in the caption of Figure 2

24. Line 243: Suggest editing “²³⁰Th-U and ²³⁴U-U age dates” to “²³⁰Th-U and ²³⁴U-U ages.”

Changed

25. Line 244: “²³⁴U age” should be edited to “²³⁴U-U age.”

Changed

26. Line 251: Suggest adding rate of decline to the dashed line. Some of the sample symbols are not as clear as the others, it is unclear why but perhaps they are not all on the same layer within the figure, please make sure all are easily legible (one glaring example is the sample at +3.2 m, 500 ka). May be useful to add a small visual key showing what the dense blue, white, and intermediate colors mean rather than stating it in caption; would make the figure easier to grasp at first glance. Are the black horizontal lines through the data symbols intended to show age uncertainty? If so, please state this in the figure caption, and whether this is a 1 or 2 sigma uncertainty.

Updated the layers to make sure they are all legible.

We did not add additional text to the figure itself (regarding both the dashed line and the visual key) because they are described in the caption where we find it more suitable.

Added “All age uncertainties are 2σ ” to the caption of Figure 3.

27. Line 394: Remove the space between “ \pm ” and “15.”

Removed

28. Line 417: The decrease in water-table is quoted as being from 573 to 584 ka, it seems like this should be inverted to move forward in time to present, and not backwards.

This statement is correct. Within uncertainty (± 14 ka) it is forward in time.

We added:

“(within uncertainty this is forward in time)”

29. Line 432: The comma is missing after “i.e.”

Added

30. Line 466: It could be helpful to the reader to annotate Panel D with arrows indicating direction of strong/weak monsoon.

Added an arrow showing weak monsoon associated with less negative $\delta^{18}\text{O}$ values (Figure 5).

31. Line 467: It is noted that Dataset B is on an inverted Y axis, however Datasets A and C are both also on inverted Y axes. Potentially worth noting this too.

Added “Note the inverted y-axis to better recognize the trends” as a general remark to this caption and removed it from Dataset B.

32. Line 469: Dataset D is described as “the Asian monsoon” which is inaccurate. Please adjust this to be an accurate description, like “ $\delta^{18}\text{O}$ record used as proxy for the Asian monsoon.”

Left unchanged because it is clear from the context and is a commonly used abbreviation. If changed here, we would need to change also e.g. “global sea level” and in the main text making it less readable.

Supplemental Material:

33. Line 39: The main text uses the notation “Devils Hole #2” while the notation “Devils Hole II” is used here. Please align these.

Corrected to “#2” everywhere.

34. Line 53: The main text uses the notation “Devils Hole #2” while the notation “Devils Hole II” is used here. Please align these.

Corrected to “#2” everywhere.

35. Line 65: Change to “Figure S2” as Sample G-426 is shown in Figure S2, not Figure S1.

Changed accordingly.

36. Line 69: Figure S1 - state that these images do not show the top of all cores, e.g., Cores H and I-bis. Is there a reason for this, are these images published elsewhere? If so, it would be helpful to state the reason / reference relevant publication.

Shown are all segments relevant for this publication. Other parts are beyond the scope of this study and do not influence the results or interpretation.

Added to the caption of both Figs. S1 and S2:

Shown are continuous segments used in this study. Top and bottom parts of cores not used in this study are not shown.

37. Line 72; Change “I:” to “I-bis:” as the core image in this figure is for Core I-bis.

Corrected

38. Line 74: Figure S2 - state that these images do not show the top of any of the cores. Is there a reason for this, are these images published elsewhere? If so, it would be helpful to state the reason / reference relevant publication.

See comment 36 above.

39. Line 98: Figure S5 - the left overview scale bar labeled “1 mm” does not make sense given the zoomed in microscope scale bar, I believe that the overview scale bar should be labeled “1 cm” so please check and adjust accordingly.

Yes, this should be cm. Changed.

40. Line 112: The “:” is missing after the Figure S8 title.

Added

41. Line 113: Capital letter missing from “figure 3.” Perhaps the “Note” that is found at the bottom of Figure S9 would be useful to add to this figure caption too as the individual uncertainties are also difficult to read.

Capitalized “Figure”.

Added “Uncertainties of individual data points older than 450 ka are difficult to read, but they are all in a narrow range between ± 14 ka and ± 17 ka” to Figure S8.

42. Line 115: The “:” is missing after the Figure S9 title. Capital letter missing from “figure 3.”

Added “:”; capitalized “Figure”.

43. Line 116: Capital letter missing from “figure S8.”

Changed

U-Th Data Set:

44. Description Tab, Line 6: Correct the spelling of “sumbergence”; the b and m are inverted.

corrected

45. Description Tab, Line 18: Correct “V&X” to “W&X.”

corrected

Reviewer #2 (Remarks to the Author):

I greatly enjoyed reading this manuscript, which builds on the earlier work of Wendt et al. (2018) and Szabo et al. (1988) to piece together the groundwater history of (part of) the Great Basin USA using Devils Hole vein calcites over the last 800-odd kyr. The data presented is very impressive indeed, and I cannot think of any places worldwide where such a detailed chronology of long-term aquifer recharge history has been assembled. Due to the chronology, the authors are also able to place their reconstruction in the context of orbital forcing and paleoclimate. They also provide new insights on the region's tectonic history. Therefore, this paper will attract wide interest in the Earth Science community and is potentially suitable for publication in COMMSENV.

The paper is well illustrated and the quality of the writing is outstanding (this makes a refreshing change!), with only a few minor edits marked on the Word docs. I also appreciated the level of detail provided in the Table S2. The supplementary information is very useful - it is great to see the cores and thin-section images, which will be appreciated by many in the speleothem community.

Regarding the comments in the two attached files:

We incorporated all suggested changes from the commented main file and supplementary file. Apart from minor upper/lower cases, missing “,” and minor changes in wording, we introduced the usage of “kyr” instead of “ka” when referring to durations and not a point in time. Also, the part about outliers was shortened (third paragraph in the results section). We did not discontinue the usage of the term “long-term component of decline of the water table” because we think it is important to highlight that this is a superimposed effect.

My main criticism relates to the narrative, or pitch, of the ms. This is immediately evident in the abstract, which, although very informative, neither identifies nor contextualises the wider scientific problem being addressed by the paper. This would seem out of alignment with the requirement for most Nature journals (including this one, I assume?). The issue somewhat continues into the Introduction, where there is background and rationale provided (as per the journal guidelines) but the background is almost exclusively related to previous studies at the site/in the region (which is necessary) with little by way of background or context to the wider scientific problem. For example, the content of the first part of paragraph #2 could be expanded to become the lead paragraph of the Introduction section that exposes the wider/bigger issue being addressed, after which comes the brief site info/previous research and paves the way for this new work. I guess what I am suggesting is that perhaps the ms should be tweaked to make it more suitable to COMMSENV. As it stands, in spite of the excellent data/results, the ms appears more appropriate for EPSL or Geology (but it easily has the potential to be in this journal because of the quality and uniqueness of the data).

We reworked the abstract and introduction to better contextualize our study and to highlight the wider scientific question. Discontinuous climate archives and dating limitations are key limitations in our understanding of climate change mechanisms. This study provides a significant step forward.

Reviewer #3 (Remarks to the Author):

The manuscript provides a new and extended Devil's Hole record covering over 750,000 years of groundwater variability. The authors conduct meticulous work on mammillary calcite and folia formations to derive robust ages to constrain groundwater variability well outside the limits of traditional U-Th ages, ~600,000 years. In their discussion, the authors try and associate groundwater variability within Devils Hole/ d234U_i to tectonic and climate processes. The manuscript will be valuable from a geochronological, paleoclimatological, paleohydrology, as well as tectonics perspective. The publication of the manuscript I believe will further encourage conversations and novel research between fields such as karst hydrology, speleothem sciences, paleoclimatology, and sedimentology, which is required to solve signals from this complex system. I enjoyed reading the manuscript though I think addressing certain statements and assumptions (see attached document) will further strengthen the paper.

I suggest minor revisions and upon revision would be supportive of this paper's publication.

The attached document is copied here below and answered in detail:

In-Depth Review of Steidle et al., 2023 entitled 'Moisture availability and groundwater recharge paced by orbital forcing over the past 750,000 years in the southwestern USA'

The following review is written in chronological order with the aim of strengthening the arguments and addressing uncertainties or lack thereof within the manuscript.

The Abstract is clear and concise and communicates the main results and conclusions. I would recommend that in L28, the authors state by providing a numerical number or range for the 'older half of the record' where the long-term decline is posited.

Changed to "prior to 438,000±14,000 years ago"

L44-45, it is recommended that authors use updated and more recent references that detail extensive work being done in southwestern US with respect to groundwater variability:

Thank you for this suggestion (see individual comments below). Although somewhat old, the Deacon citation deals with the study site specifically and is kept.

Miller, O.L., Putman, A.L., Alder, J., Miller, M., Jones, D.K. and Wise, D.R., 2021. Changing climate drives future streamflow declines and challenges in meeting water demand across the southwestern United States. *Journal of Hydrology X*, 11, p.100074.

Was added

Masbruch, M.D., Rumsey, C.A., Gangopadhyay, S., Susong, D.D. and Pruitt, T., 2016. Analyses of infrequent (quasi-decadal) large groundwater recharge events in the northern Great Basin: Their importance for groundwater availability, use, and management. *Water Resources Research*, 52(10), pp.7819-7836.

We find this paper to be a bit outside of the scope here and did not include it.

MacDonald, G.M., 2010. Water, climate change, and sustainability in the southwest. Proceedings of the National Academy of Sciences, 107(50), pp.21256-21262.

MacDonald was already used but cited with a typo. It is corrected now.

Meixner, T., Manning, A.H., Stonestrom, D.A., Allen, D.M., Ajami, H., Blasch, K.W., Brookfield, A.E., Castro, C.L., Clark, J.F., Gochis, D.J. and Flint, A.L., 2016. Implications of projected climate change for groundwater recharge in the western United States. Journal of Hydrology, 534, pp.124-138.

Was added.

Siirila-Woodburn, E.R., Rhoades, A.M., Hatchett, B.J., Huning, L.S., Szinai, J., Tague, C., Nico, P.S., Feldman, D.R., Jones, A.D., Collins, W.D. and Kaatz, L., 2021. A low-to-no snow future and its impacts on water resources in the western United States. Nature Reviews Earth & Environment, 2(11), pp.800-819.

Was added.

Figure 1A: It might be helpful to conceptually draw the major drivers of rainfall over Devils Hole in the modern or during periods of large Ice Sheets such as the Laurentide/Cordilleran over the US. Authors cite papers from Bhattacharya and Oster, which provide conceptual diagrams, during their discussion in the southernly shift of the moisture trajectories.

Although this is important background information, it is not in the focus of this paper.

L108-110; stating that ‘the groundwater table shows the same elevation in both caves, suggesting that they are hydraulically connected’ warrants a citation/ presentation of the data. I might have missed that if it is presented in the manuscript.

We surveyed both Devils Hole proper and Devils Hole #2 and the difference in water-table elevation was found to be less than 8 cm, which is well within the uncertainty of the applied technique (DistoX). Added to the manuscript:

The groundwater table elevation is the same in both caves (based on our own survey), suggesting that they are hydraulically connected.

L141-149: It would be helpful to the reader to reference Table S1 here. It is not clear to me why, mammillary calcite <10mm is considered thin and associated with a brief increase in water table? 10mm is thicker than 5mm, which is being associated here as a decrease in water level. Wendt et al., 2018 in their ‘Interpretation of thin mammillary calcite layers’ section discuss ‘Mammillary calcite (subaqueously deposited) intervals (≤ 0.5 mm in width)’

The supplementary material is referenced in the sentence before where the different types of deposit are discussed. Added there:

(see Supplementary material section “Types of calcite deposits and their relationship to the water table” and table S1 for additional information on folia and its subtype referred to as proto-folia).

Mammillary calcite less than 5 mm away from folia is considered to represent an increase/decrease in water table. If a sample was taken in the middle of a <10 mm thin mammillary calcite layer bracketed by folia, the sample position is <5 mm away from both sides of the folia and hence represents both an increase and decrease of the water table (called “brief rise” then). Added to the manuscript:

Samples of thin (<10 mm) mammillary calcite layers bracketed by folia (i.e., <5 mm above prior folia and <5 mm below subsequent folia) are attributed to brief rises of the water table.

The methodology is consistent with Wendt et al. (2018), citing from the paper:

“In addition, 13 of 72 boundaries included a calcite layer that was approximately 1 mm in width before folia deposition resumed. We interpreted these thin layers as a brief increase in the water table. A single date was measured in these cases, and the reported brief increase in water table elevation was centered on the measured date (fig. S1).”

Figure 2 and associated text in L172-196, might help to underline the difference between the 53 230Th-U and 86 234U-U. IF there are a total of 92 samples that were analyzed and 5 were out of stratigraphic order, which gives 87 samples, what is the sampling distribution of the 86 ages.

See also our answer to major comment 3 of reviewer #1. In brief: we added a half-page description in the supplementary material and made minor changes in the main text.

L200 authors use IDs of the samples but in other places, they interchange with the depths associated with the cores, for example, L214. It might be helpful to be consistent. As the focus of the manuscript is associated with water level depths, I would suggest that the authors use numerical values and add in parentheses the core IDs.

Sample IDs are used only at a few places where it is necessary to point to specific samples and the depth information alone is not sufficient to explain the reasoning. However, sample elevation is always given. We wrote: “in four cores between +4.6 m and +8.3 m”.

L200, it is unclear if the oldest mamillary calcite date of 820 ka is the average taken from the 5 cores, Ibis825, N771, N775, J992, and G867. If it is an average of the cores, it would be helpful to mention that and assign an uncertainty. Similarly, it is stated in L201, ‘750 ka is present at all sampled elevations above 4m.’ It is unclear whether this is an average or a single date from a single core.

We added an uncertainty description and made this point clearer. We expanded the first sentence of the paragraph:

“The oldest mammillary calcite deposition is recorded in four cores between +4.6 m and +8.3

m elevation and dates to 820 ka. This value is the mean of samples Ibis825, N771, N775, J992 and G867. Its uncertainty is limited by the ^{234}U -U age model and is about ± 15 ka.”

L212 missing word after older ‘than’

Changed

The following line states the cores but, in this case, only mentions the heights, which is more important, I agree, but to be consistent with L200, it would help to add core IDs/

See comment about L200 above.

L219: I might have missed why <100 ka is considered shorter? Similarly, maybe provide a line of justification on using >100 ka as ‘long’

A mean water-table drop over 280 ka was reported in this sentence. 100 ka is an arbitrary threshold to make the point that faster changes (due to a different driving mechanism) are superimposed on this long (and steady) trend.

Given that the other reviewers did not comment on this we did not change this sentence.

L232: Should Figure S2 instead be S3 as this is the first petrographic image depiction the transition mentioned? Further, it might be helpful to include the petrographic figures more robustly in the main text.

Double checked the figure references. They are correct (not changed).

We find discussing petrographic details in the main text to distract from the main topic (the water table).

Next, Fluid inclusions shown in Figure S3 and S6 are not really discussed in the manuscript and hence I suggest either include them in the discussion or remove them.

Fluid inclusions are relevant to justify our classification of the 3 described types of calcite deposits discussed in the supplementary material. This is important information to support our classification documenting a decrease/increase, brief rise or submergence, but as stated in the last comment, this is not the main focus of our study. We believe this information should be kept in the supplementary material.

L234: When calculating average growth rates and then comparing them with Li et al., 2020, I would recommend adding error bars. Similar with the growth rate calculated in L237.

The growth rate is the result of a line fit through 2 points (start and end age). The numeric uncertainty is small ($<10\%$) and only defined by the age uncertainties of the start and end point. However, this number would imply a degree of certainty that is not true. The main

criterion here (also discussed in the text) is whether or not it is continuous and that can only be addressed by petrographic investigations. No changes made.

Figure 3: I appreciate the effort in constructing a hypothesized visual representation of the water table. I would recommend that the authors provide an explanation on what “submerged with high certainty” L254 means. Is that at the 95% CL? How was that calculated? Similarly, provide an explanation for “not submerged with high certainty.” Lastly, how was the qualitative measurement done? Was it an interpolation? Adding this to Figure S7 might help in understanding how the shading was developed.

The blue shading in Figure 3 was hand drawn to make the figure more readable (as stated in the manuscript: “visual aid”). There is no mathematical calculation behind it otherwise it would have been stated. Figure S7 is independent from that.

Further, it is interesting to note that the dashed black line highlighting the decline and discussed is evident by only periods of brief (depicted by the diamonds) rise of the water table around 650 and 750 kyrs. What is temporal range constituting this ‘brief’ period? Is it less than 10 kyrs? But then, authors state in L268-269 that timescales smaller than 10 ka are not resolvable due to dating uncertainty. So I am a little confused here...

We added to the caption:

The black dashed line is the long-term component of decline discussed in the subsection “Long-term water-table decline component”.

We do not fully understand the comment about the 10 kyr but believe it may have been clarified by pointing to the relevant section for the dashed line.

Further, the water table with a large number of data points is around 10 m covering 400-750 kyrs. So, I was curious if the authors have thought whether a sampling bias produces the suggested decline?

A sampling bias as discussed in many (vadose) speleothem studies appears unlikely due to the proven continuous deposition throughout the whole last 750 kyr. We are convinced that there is a signal (discussed in length in the section “Long-term component of water table decline”).

We added to the end of the first paragraph of this section:

“A sampling bias to explain this is considered unlikely because continuous deposition is documented for the last 750 kyr.”

L28-281: It might be helpful to briefly state what the ‘other processes’ are and how they are indicative of being negligible to water table elevation.

We added:

Wendt et al. (2020) discussed different sources of $\delta^{234}\text{U}_{\text{initial}}$ variations in Devils Hole and found processes other than the discussed effect of water-table elevation (such as changes in the groundwater source, flow rate, or flow path) to be negligible over the past 350 kyr.

L286: How was the significance identified in the following, ‘only one significant period’? It would also help to identify in Figure 4 by a vertical dashed line when these changes occur.

The following was added:

In the time period between 820 ka and 350 ka only one significant (i.e., uncertainty ranges of data points in the minimum do not overlap with the uncertainty range in the following maximum) interval of low $\delta^{234}\text{U}_{\text{initial}}$ dated to around 410 ka was followed by an increase in $\delta^{234}\text{U}_{\text{initial}}$ (Figure 4).

L288: Could you add a numerical value when ‘lower amplitude’ is mentioned? Similarly, what is meant by uncertainties larger than the amplitude? Lastly, indicate again Figure 4 when these changes occur.

Changed to the following (for the figure see comment above):

Similar rises in $\delta^{234}\text{U}_{\text{initial}}$ but smaller in amplitude than twice the uncertainty of individual data points occurred around 700 ka and 630 ka.

L293: ‘Major changes to the aquifer,’ such as...

Changed to:

This could indicate that there was no long-lasting lowstand during this period, but given a data gap of about 70 kyr, major changes of the water-table elevation cannot be excluded.

L309: It is interesting to notice d18O reflected close to glacial values. What are the possible reasons for d13C behaving so differently? A line or two of discussion might be helpful.

They both reflect glacial values. An unfortunate line break separating the “-“ from the “1.7‰ in d13C” may have caused irritation.

L355: ‘further past’ reads incorrectly.

The wording of another reviewer was adopted and this term was removed.

L388: from should be for

“The Devils Hole data document a strong decline in water table between 438 ± 14 ka and 403 ± 6 ka from more than +9.9 m to less than +3.2 m, associated with Termination V, as well as a minimum in $\delta^{234}\text{U}_{\text{initial}}$ as expected from a major groundwater lowstand (Wendt et al., 2019).”

The wording is correct (“more than +9.9 m” was the (absolute) water table elevation at 438 ka).

L389: Wend et al., 2019 is not cited.

Corrected to Wendt et al. (2020)

L409: from should be ‘for’

“Termination VII is reflected in a water-table decline between 660 ± 15 ka and 627 ± 15 ka from more than 18.7 m to just below 6.8 m.”

This sentence is correct.

L409 and L417: The authors should address the result that terminations are within error of each other; 660 15 and 627 15 kya. Separately, 573 14 and 584 14 kya.

These are the start and end dates of the termination. We do not understand the statement “terminations are within error of each other”. This is not the case. Termination VII (which lasted from 660 ka to 627 ka) is not within error of Termination VIIa (from 573 ka to 584 ka, considering the uncertainties.

Further, Termination VIII brief rises and the lowstand is within the <10 ka age uncertainty. Therefore, I am not sure how much weight can be applied to these interpretations.

“Termination VIII is reflected by a decline after brief rises at 19.5 m at 719 ± 14 ka and at 18.7 m at 724 ± 15 ka and a lowstand at 15.8 m after 720 ± 14 ka.”

We assume this comment refers to this sentence, but we do not fully understand the comment. All three given dates are independent evidence of the same event (i.e., a water table decline at Termination VIII) and they all agree within error. The confusion may arise from the use of ‘brief rises’ for only the fact that there is a decline following the brief rise is relevant here.

We changed the wording of to:

“Termination VIII is reflected by a decline after 719 ± 14 ka at 19.5 m, after 724 ± 15 ka at 18.7 m, and by a lowstand at 15.8 m after 720 ± 14 ka.”

L442: If authors state ‘amplitudes smaller than during most Terminations’ it would be helpful to provide approximate numerical values. Smaller compared to what?

As discussed extensively in the first subsection of the Discussion, quantifying amplitudes is difficult, because there is no upper limit to water table makers. We removed this half sentence because we cannot provide meaningful numerical values. Removed:

~~“amplitudes smaller than during most Terminations.”~~

L449: A new paleoclimate simulation might be another good citation to add:
Oster, Jessica L., et al. "North Atlantic meltwater during Heinrich Stadial 1 drives wetter climate with more atmospheric rivers in western North America." *Science Advances* 9.46 (2023): eadj2225.

We added the citation.

L452: I am curious whether the growth or decay of the Cordilleran ice sheet or the Laurentide Ice Sheet (L458) are more influential to the moisture balance for Devils Hole?

We are also curious and hope it can be answered in future studies.

Decision letter and referee reports: second round

17th May 24

Dear Mr Steidle,

Your manuscript titled "Moisture availability and groundwater recharge paced by orbital forcing over the past 750,000 years in the southwestern USA" has now been seen by our reviewers, whose comments appear below. In light of their advice we are delighted to say that we are happy, in principle, to publish a suitably revised version in Communications Earth & Environment under the open access CC BY license (Creative Commons Attribution v4.0 International License).

We therefore invite you to revise your paper one last time to address the remaining concerns of our reviewers. At the same time we ask that you edit your manuscript to comply with our format requirements and to maximise the accessibility and therefore the impact of your work.

EDITORIAL REQUESTS:

*****Please take care to match our formatting and policy requirements. We will check revised manuscript and return manuscripts that do not comply. Such requests will lead to delays. *****

SUBMISSION INFORMATION:

In order to accept your paper, we require the files listed at the end of the Editorial Requests Table; the list of required files is also available at <https://www.nature.com/documents/commsj-file-checklist.pdf> .

OPEN ACCESS:

Communications Earth & Environment is a fully open access journal. Articles are made freely accessible on publication under a CC BY license (Creative Commons Attribution 4.0 International License). This license allows maximum dissemination and re-use of open access materials and is preferred by many research funding bodies.

For further information about article processing charges, open access funding, and advice and support from Nature Research, please visit <https://www.nature.com/commsenv/article-processing-charges>

At acceptance, you will be provided with instructions for completing this CC BY license on behalf of all authors. This grants us the necessary permissions to publish your paper. Additionally, you will be asked to declare that all required third party permissions have been obtained, and to provide billing information in order to pay the article-processing charge (APC).

[redacted]

Best regards,

Carolina Ortiz Guerrero, Ph.D.
Associate Editor
Communications Earth & Environment

REVIEWERS' COMMENTS:

Reviewer #1 (Remarks to the Author):

The revised manuscript has responded to all my major and minor comments, either directly addressing them in the manuscript or responding to them through rebuttal. A significant portion of text was added to the main manuscript to expand on assumptions related to $\delta^{234}\text{U}$ initial estimates and processes that can affect $\delta^{234}\text{U}$ initial. My previous comments on this point still stand, and I have concerns with extrapolating $\delta^{234}\text{U}$ initial values 160 kyr beyond the calibration dataset with no inflation of uncertainty and thereby producing ^{234}U - ^{238}U ages with potentially underestimated uncertainties. However, these assumptions have now been more clearly laid out in the text and can be more easily interrogated by readers. Detailed text was also added to the Supplemental Material carefully explaining inclusion/exclusion of U-series data used for analysis and figures, which has added much-needed clarity. The authors also made substantial edits to multiple figures based on my comments, particularly to Figure 2 adding a more robust statistical examination of the data comparison and a fairer presentation of the data. This revised manuscript, presenting an important groundwater-table reconstruction with a well-executed example of a unique approach to extending U-Th dating beyond traditional timescales, can now be recommended for publication in Nature Communications Earth and Environment.

Reviewer #2 (Remarks to the Author):

I have been through the revised ms and the rebuttal documents, and I am satisfied with the authors' responses to both my concerns and those of the other two reviewers.

Reviewer #3 (Remarks to the Author):

The authors have addressed all the comments posed to them and incorporated relevant changes in the manuscript. I look forward to seeing the paper in print and congratulations to all co-authors on their effort.